# Trajectory-modulated hippocampal neurons persist throughout memory-guided navigation

Nathaniel R. Kinsky [1,2✉], William Mau [1,3], David W. Sullivan[1], Samuel J. Levy[1,4], Evan A. Ruesch[1] & Michael E. Hasselmo [1✉]

Trajectory-dependent splitter neurons in the hippocampus encode information about a rodent's prior trajectory during performance of a continuous alternation task. As such, they provide valuable information for supporting memory-guided behavior. Here, we employed single-photon calcium imaging in freely moving mice to investigate the emergence and fate of trajectory-dependent activity through learning and mastery of a continuous spatial alternation task. In agreement with others, the quality of trajectory-dependent information in hippocampal neurons correlated with task performance. We thus hypothesized that, due to their utility, splitter neurons would exhibit heightened stability. We find that splitter neurons were more likely to remain active and retained more consistent spatial information across multiple days than other neurons. Furthermore, we find that both splitter neurons and place cells emerged rapidly and maintained stable trajectory-dependent/spatial activity thereafter. Our results suggest that neurons with useful functional coding exhibit heightened stability to support memory guided behavior.

[1] Center for Systems Neuroscience, Boston University, 610 Commonwealth Ave, 7th Floor, Boston, MA 02215, USA. [2] Department of Anesthesiology, University of Michigan, 1301 Catherine St. Rm 7433, Ann Arbor, MI 48109, USA. [3] Icahn School of Medicine at Mount Sinai, 1470 Madison Ave, 10th Floor, New York, NY 10029, USA. [4] Graduate Program for Neuroscience, Boston University, Boston, MA, USA. ✉email: nkinsky@gmail.com; hasselmo@bu.edu

Place cells in the hippocampus encode the current position of many different animals and humans[1–4] supporting the known role of the hippocampus in spatial memory and navigation across species[5]. However, the hippocampus is also widely known for its role in supporting the encoding, retrieval, and consolidation of non-spatial long-term memories[6], suggesting that it must represent variables beyond an animal's current location. Indeed, recent studies have demonstrated that the hippocampus encodes the dimensions of a given task, from odors[7,8] to time[9–13] to tones[14]. One early demonstration that the hippocampus encodes dimensions beyond an animal's current location was the discovery of trajectory-dependent neurons or splitter neurons[15,16], cells whose firing rate within a particular position were modulated based on the animal's past or future trajectory in a spatial alternation task, referred to hereafter as trajectory-dependent activity or trajectory coding. The generation of this neural correlate suggests a potential mechanism allowing the hippocampal code to support both memory and decision-based planning.

Several studies have demonstrated that place cell firing fields move, or remap, their locations in response to new learning during a spatial memory task[17,18] highlighting that the flexible adjustment of place field locations is important for learning new information. Conversely, the ability of hippocampal neurons to maintain the same firing location in the absence of learning might support long-term memory retrieval. In support of this idea, a recent study illustrated that neurons with place fields located near a hidden goal were more stable over time than cells with fields in other locations[19]. Two other experiments found that increasing rodents' attention to a task selectively heightened stability in neurons that encoded task-relevant features[8,20]. These studies, along with the finding that place cells with fields in close proximity to a goal location exhibit heightened activity in post-learning sleep[18], suggest that the utility of a neuron's information to task performance influences its long-term stability.

Thus, since splitter neurons provide immediately relevant information for performing a spatial alternation task, we hypothesized that these neurons are important for successful task performance. Furthermore, we hypothesized that due to their utility, splitters may be preferentially stabilized when compared to place cells. Specifically, we addressed three lines of inquiry. First, how does the level of trajectory-dependent information within the hippocampus correlate with behavioral performance? Second, given the steady evolution of hippocampus activity patterns across days[21–24], do splitter neurons remain part of the active population longer than other cells, thus providing a longer lasting memory or planning signal to guide behavior? Third, once a neuron establishes trajectory-dependent activity, is it less prone to remapping than other neurons? These questions are particularly relevant since trajectory-dependent activity has been observed in other tasks[25–27] and could be employed more generally by the hippocampus to guide the appropriate behavior based on environmental cues[28].

To track neurons across long timescales, we paired a continuous spatial alternation task with in vivo miniscope recordings of GCaMP6f activity in dorsal CA1 of freely moving mice[22,29]. This technology allowed us to not only track the long-term activity of neurons, but also to adequately characterize the heterogeneity of trajectory-dependent activity in the hippocampus, since we can simultaneously record from a large number of neurons in each session. First, we found that some attributes of trajectory-dependent coding correlate with task performance. Second, we established that a neuron's functional coding properties, or information content, are important for predicting its long-term activity. Splitter neurons are more likely to be persistently active in subsequent sessions than return arm place cells

and non-place cells indicating that neurons which provide more adaptive information contribute longer lasting input to downstream structures. Third, we found that trajectory-dependent neurons display more consistent long-term information about an animal's location than pure place cells. Fourth, we found that the population as a whole undergoes a rapid onset of trajectory-dependent activity followed by stable trajectory coding thereafter. Last, we discovered that recruitment of context-dependent splitter cells peaks several days into training, whereas place cell recruitment peaks on the first day. These results combined suggest that neurons that with behaviorally relevant coding properties exhibit high short and long-term stability, which could enable them to more consistently and effectively support memory-guided behavior. Our research paves the way for future studies investigating how heterogeneity in the neural code might support acquisition and retention of more complex behavioral tasks.

## Results

**Behavior and imaging.** Food deprived mice ($n = 4$) with neurons expressing GCaMP6f in region CA1 of the dorsal hippocampus were trained to perform a continuous spatial alternation task on a figure-8 maze (Fig. 1a) while we simultaneously recorded calcium activity using a miniaturized microscope. Mice exhibited a range of learning rates, taking from 5 to 21 sessions to acquire the task, which was defined as the third consecutive session of performance at or above our criteria of 70% correct (Fig. 1b). Mice performed continuous alternation at or greater than criteria on average throughout the course of the experiment (Fig. 1c). We utilized custom-written software[24,30] to extract neuron ROIs (Fig. 1d), construct their corresponding calcium traces, and identify each ROI's putative spiking activity (Fig. 1e, Supplementary Fig. 1c, d). Using this technique, we recorded from large numbers of neurons (243–1205 neurons per ~30 min session) and successfully tracked them across days by comparing the distance between neuron ROI centroids (Supplementary Fig. 1a) and verifying that ROIs did not change orientation of their major elliptical axis between sessions (Supplementary Fig. 1b). We observed no systematic changes in calcium trace kinetics or fluorescence across sessions, indicating stable levels of GCaMP expression (Supplementary Fig. 1e, f). Nevertheless, we excluded any potentially unhealthy neurons that had half-decay times >2 s (13% ± 6.7% of neurons across all sessions, mean ± std., $n = 68$ sessions) from further analysis.

**Trajectory-dependent activity is maintained across days.** The initial studies by Frank et. al and Wood et al. used electrophysiology in rats to establish the existence of trajectory-dependent splitter cells in the hippocampus[15,16]. Thus, we first wondered if we could detect trajectory coding in a different species while using a technique with much lower temporal resolution. To do so, we constructed tuning curves representing the probability that a given neuron had calcium activity at each spatial bin (1 cm) along the stem in correct trials only, and classified neurons as trajectory-dependent splitters if at least three bins displayed a significant difference between their tuning curves ($p < 0.05$, permutation test). We found that we were capable of not only identifying trajectory-dependent cells on a given day (60 ± 23 neurons, mean ± s.e.m. across all four mice), but that in many cases these neurons maintained significant trajectory-dependent activity across multiple days (Fig. 2a, b, Supplementary Fig. 2c–n). Significant trajectory-dependent activity was exhibited by 10% of neurons active on the maze stem across all sessions (12%, 5%, 12%, and 9% for individual mice); note that this method for identifying trajectory-dependent activity is more conservative than that used in previous studies[15,16,31]. Apparent trajectory-dependent activity could also potentially result from factors such as systematic

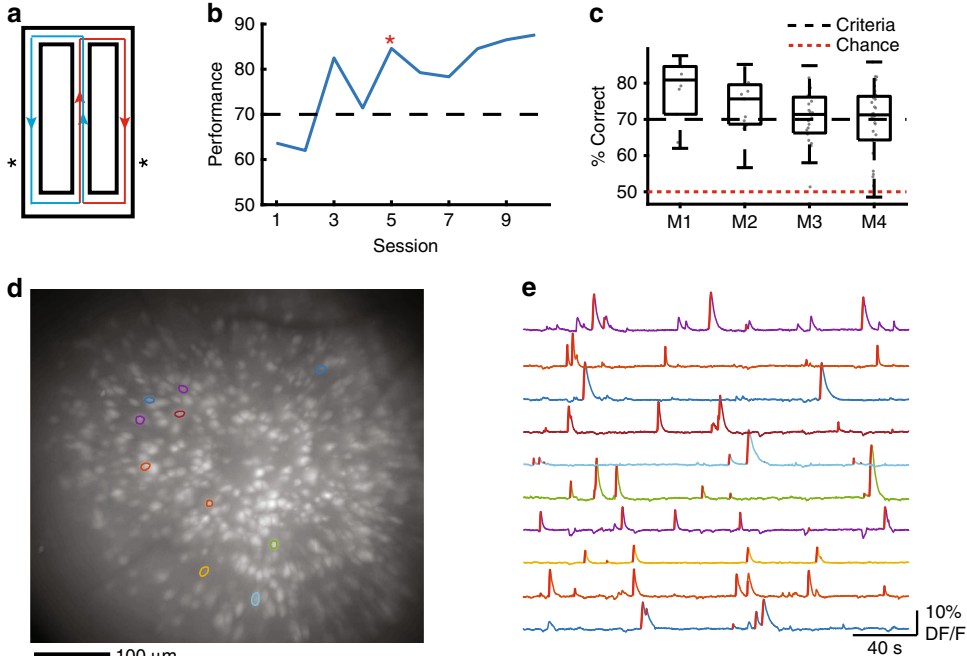

**Fig. 1 Experimental setup and imaging. a** Alternation Maze. Blue = Left turn trajectories, Red = Right turn trajectories, *= location of food reward. **b** Example learning curve for one mouse. Black dashed = acquisition criterion (70%), red asterisk = task acquisition day. **c** Performance summary for all four mice ($n$ = 10, 10, 23, 28 sessions). Black dashed = criterion, red dashed = chance. Box plots show median and 25th/75th percentiles, whiskers show data extent excluding outliers. **d** Representative maximum projection from one imaging session (873 neurons detected) with 10 neuron ROIs overlaid. The maximum projection was consistent for each mouse across 10, 7, 23, and 28 sessions with 748 ± 102, 337 ± 46.5, 1030 ± 79.7, and 611 ± 147 neurons detected respectively (mean ± std). **e** Example calcium traces for ROIs depicted in (**d**). Red lines on the ascending phase of each calcium event indicate inferred spiking activity.

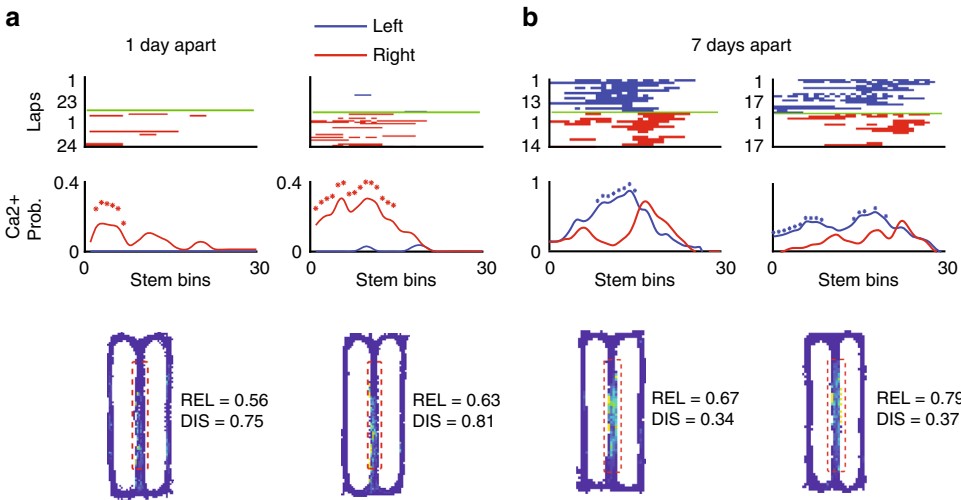

**Fig. 2 Trajectory-dependent activity persists across days. a** (top) Calcium event rasters along the stem for correct trials for two sessions recorded one day apart, sorted by turn direction at the end of the stem. Blue = left turns, Red = right turns. (middle) Calcium event probability curves for each turn direction. *indicates bins exhibiting significant trajectory-dependent activity ($p < 0.05$, one-sided shuffle-test). Curves were smoothed using a smoothing spline for visualization purposes ($p = 0.9$, see Methods). (bottom) occupancy normalized calcium event map with reliability (REL) and discriminability (DIS) scores shown. Blue = no calcium activity, yellow = maximum calcium activity, red dashed = extent of stem considered in above plots. **b** Same as a, but for a different mouse and for sessions 7 days apart.

variations in the mouse's lateral position along the stem. We addressed this in two ways. First, we limited the portion of the maze we considered the stem to exclude any areas where the mouse exhibited stereotypical turning behavior by eye (Fig. 2a, b, bottom). Second, we performed an ANOVA for each splitter neuron which included the animal's upcoming trajectory, position along the stem, speed, and lateral position along the stem as covariates[16]. We found

that a high proportion of our splitter neurons were significantly modulated by upcoming turn direction after accounting for speed and lateral stem position (89%, 72%, 76%, and 83% for individual mice). Neurons that did not exhibit turn direction modulation after accounting for speed and lateral position were not categorized as splitter neurons in subsequent analyses. Together, these results indicate that trajectory-dependent coding exists in mouse CA1 and

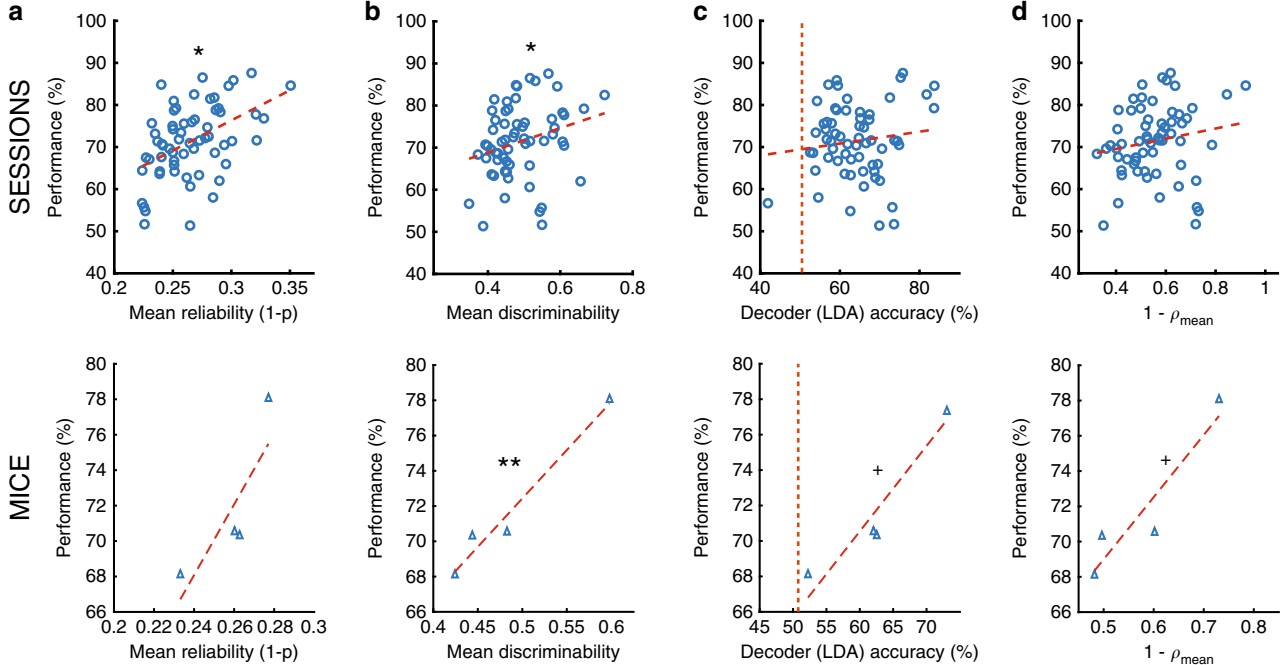

**Fig. 3 The quality of trajectory-dependent activity correlates with performance. a** (top) Performance for each session versus the mean reliability value for all cells active on the stem from that session. Circles = all sessions, all mice. *$\rho = 0.47$, $p = 9.9 \times 10^{-5}$ Pearson correlation. (bottom) Same as top but for each mouse, triangles = average for each mouse. **b** Same as a, but for the mean discriminability value. *$\rho = 0.26$, $p = 0.039$, **$\rho = 0.97$, $p = 0.017$ Pearson correlation. **c** Same as (**a**), but for the mean LDA decoder accuracy. Dashed orange line indicates chance-level decoder accuracy assessed by shuffling trial identity. $^{+}\rho = 0.89$, $p = 0.057$, Pearson correlation. **d** Same as a, but for the mean value of 1—Spearman's $\rho$ between left and right tuning curves. $^{+}\rho = 0.87$, $p = 0.066$, Pearson correlation.

in many cases maintains the same activity profile across both short and long timescales. To the best of our knowledge, this is the first demonstration of hippocampal trajectory-dependent activity using calcium imaging in mice.

Additionally, we observed that the mean location of spatial firing along the stem progressed backward within each recording session such that calcium activity occurred at earlier and earlier portions of the stem with time (Supplementary Fig. 2a). This is consistent with a study reporting the backwards-migration of spatial firing with experience[32]. Interestingly, we did not find any evidence of consistent migration of spatial firing locations between sessions (Supplementary Fig. 2b).

**Trajectory-dependent activity correlates with performance.** Trajectory-dependent neurons provide information vital to task performance that might be utilized by downstream structures to inform proper motor actions[33–35]. This idea is supported by studies finding that trajectory-dependent activity markedly diminished during error trials[25,27]. Thus, we predicted that successful task performance would be associated with prominent trajectory-dependent information in the neural code of neurons active on the stem. We utilized two metrics to measure different attributes of trajectory-dependent activity: (1) reliability, which measured the consistency of a cell to fire on its preferred trial type along the entire stem, and (2) discriminability, which measured the magnitude of difference between left and right turn tuning curves along the entire stem (Methods). While most splitter neurons generally had high reliability and discriminability values, neurons with sparser calcium activity for one turn direction could exhibit low reliability and high discriminability (Fig. 2a). Conversely, splitter neurons that reliably increased their event rate for one turn direction but still exhibited activity for the other turn direction could have high reliability but low discriminability (Fig. 2b). We also assessed population-level trajectory-dependent

information by training a linear discriminant analysis (LDA) decoder to classify future turn direction at each spatial bin along the stem.

We found a positive correlation between all three metrics and the animal's performance (Fig. 3). These correlations were significant for reliability and discriminability, but not LDA decoder accuracy, across all sessions (Fig. 3a–c top). We also obtained positive correlations when we averaged across mice. These correlations were significant for discriminability, but not reliability, and the relationship between LDA accuracy and performance approached significance (Fig. 3a–c bottom). Last, for each cell, we correlated the left turn and right turn tuning curves and subtracted those values from 1 (1−ρ) as another metric for trajectory-dependent information. Note the conservative nature of this metric for measuring trajectory-coding: it produces low values (indicating high-trajectory-dependent information) for splitter neurons that shift their location along the stem between trial types but not for splitter neurons that modulate event rates in the same location. We found a positive correlation between performance and 1−ρ that approached significance when we averaged across mice (Fig. 3d), which was not surprising given the conservativeness of this metric. We obtained similar results when we focused on local metrics of trajectory-dependent activity rather than their average along the entire stem (Supplementary Fig. 3). Together, these results indicate that some attributes of trajectory-dependent activity might facilitate accurate task performance.

**Greater across-day stability for trajectory coding neurons.** Multiple studies have shown that hippocampal neurons exhibit significant turnover across days with fewer staying active within the same environment as time progresses[21–24,30]. However, these studies all treated the CA1 population as one homogeneous group. Thus, we wondered if splitter neurons, which exhibit task-

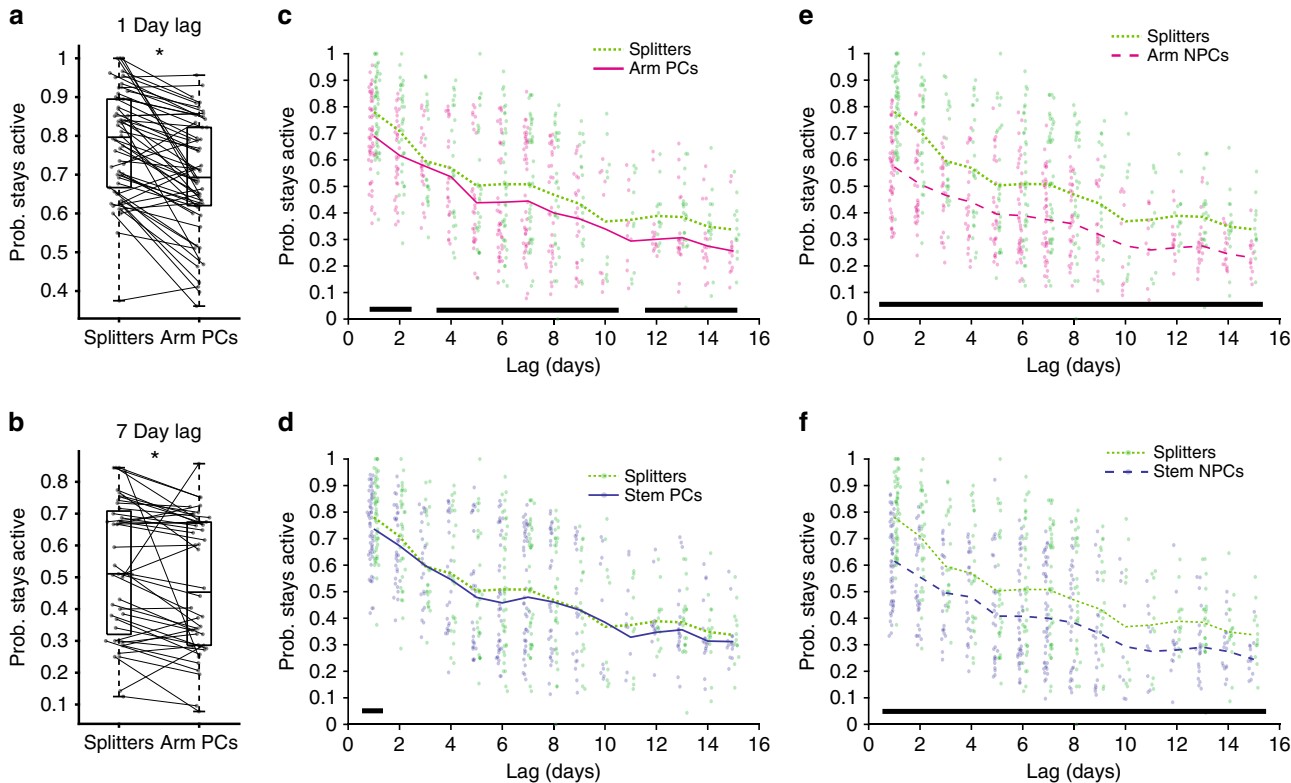

**Fig. 4 Splitter neurons are more likely to remain active between sessions than other neurons. a** Probability that splitters and arm place cells (PCs) stay active one day later for all mice. *$p = 6.7 \times 10^{-8}$, $n = 58$ session pairs, one-sided signed-rank test. Box plots show median and 1st/3rd quartiles, whiskers extend to 1.5× interquartile range. **b** Same as a, but for sessions 7 days apart. *$p = 4.1 \times 10^{-4}$, $n = 44$ session pairs, one-sided signed-rank test. **c** Probability splitters and arm place cells stay active versus lag between sessions. Dots = probabilities from individual session pairs, lines = mean probability at each time lag. Green dashed = splitters, Magenta = arm PCs. Black bars = significant differences after Holm-Bonferroni correction (15 day lags considered) of one-sided signed-rank test. See Table 1 for signed-rank test p-values at all lags, α = 0.05. **d** Same as c but for comparison to stem PCs (blue). **e, f** Same as (**c, d**) but for comparison to arm and stem non-place cells (long dashed).

relevant information, would be preferentially stabilized within the CA1 network when compared to traditional place cells. As such the hippocampus would maintain a more consistent population of neurons that could be utilized for guiding this behavior. To address this question, we first calculated the probability that each pool of neurons remained active in subsequent sessions. We found that splitter cells were more likely to remain active in a later session than arm place cells for short (Fig. 4a) and long (Fig. 4b) time lags between sessions. This higher likelihood of splitter neurons remaining active generally persisted up to 15 days later (Fig. 4c, Table 1) and was exaggerated when we compared splitters to non-place cells on both the arm and the stem of the maze (Fig. 4e, f). However, we found that splitters and stem place cells were equally likely to remain active at all time lags (Fig. 4d). This could occur because stem place cells also exhibit trajectory-dependent activity that does not quite meet our stringent splitter neuron criteria. In support of this idea, stem place cells carrying highly reliable trajectory information were more likely to remain active than those carrying relatively unreliable trajectory information (Supplementary Fig. 4g, Table 2). Alternatively, this finding could support the idea that both the animal's current and past position are relevant for task performance, which in turn could influence the stability of place cells and splitters, respectively[27]. To mitigate any sampling biases due to the higher event rate of splitter neurons (Supplementary Fig. 4a), we performed an additional analysis where we included only the most active place and non-place cells such that their mean event rate matched that of splitters. We found that splitters were generally more likely to remain active than event-rate matched non-

place cells (Supplementary Fig. 4e–f, Table 3). However, event-rate matched arm and stem place cells were equally as likely to remain active as were splitters (Supplementary Fig. 4b–d, Table 3), consistent with the idea that the stability of both place cells and trajectory-dependent cells is important for task performance[27]. We also observed no differences in stability based on the level of trajectory information in event-rate matched neurons (Supplementary Fig. 4h, Table 2). This suggests that a neuron's activity level, along with its information content, also influences whether it stays active on following days. These findings combined support the idea that the task relevance of information carried by a neuron influences its likelihood to maintain activity at later time points, which could be exploited for successful memory-guided behavior across days.

We next wondered how the information provided by splitter cells differs from that of other neurons. First, we found that firing fields were generally larger for splitters than place cells (Supplementary Fig. 5a). To further investigate, we decided to compare the long-term spatial coding properties of trajectory-dependent splitter neurons on the stem to return arm place cells (Fig. 5a). When examining spatial calcium activity over the entire map across sessions, we found that splitter neurons had significantly higher 2D event map correlation values than arm place cells (Fig. 5b) and that this effect persisted up to 15 days later (Fig. 5c) indicating that they were more stable overall. We observed even higher relative spatial stability of splitter neurons compared to stem place cells (Table 3, Supplementary Fig. 5c). The higher correlations for splitters were not explained by their place-field size nor by experimenter movements to provide reward on the return arm because stem

**Table 1 One-sided signed-rank significance values for the probability splitter vs. place cells (PCs) and non-place cells (NPCs) remain active between sessions ($n$ = session pairs).**

| Day lag (n) | 1 (58) | 2 (34) | 3 (24) | 4 (25) | 5 (30) | 6 (41) | 7 (44) | 8 (31) |
|---|---|---|---|---|---|---|---|---|
| vs. Arm PCs | 5.7e−8 | 1.1e−4 | 0.24 | 0.026 | 0.024 | 1.2e−4 | 4.1e−4 | 0.0021 |
| vs. Stem PCs | 0.0021 | 0.05 | 0.55 | 0.34 | 0.39 | 0.026 | 0.42 | 0.77 |
| vs. Arm NPCs | 2.7e−11 | 3.3e−7 | 2.3e−4 | 8.6e−5 | 3.4e−4 | 3.2e−6 | 3.2e−8 | 2.5e−4 |
| vs. Stem NPCs | 8.9e−11 | 4.9e−6 | 8.0e−4 | 9.0e−4 | 4.0e−4 | 6.0e−6 | 1.2e−6 | 8.1e−5 |
| Day lag (n) | 9 (21) | 10 (10) | 11 (10) | 12 (14) | 13 (22) | 14 (17) | 15 (13) | |
| vs. Arm PCs | 0.024 | 0.16 | 0.0067 | 9.1e−4 | 0.019 | 0.0023 | 0.024 | |
| vs. Stem PCs | 0.43 | 0.65 | 0.88 | 0.60 | 0.45 | 0.41 | 0.42 | |
| vs. Arm NPCs | 1.9e−4 | 9.8e−4 | 0.014 | 3.1e−4 | 1.2e−4 | 1.0e−3 | 6.1e−4 | |
| vs. Stem NPCs | 6.9e−4 | 0.014 | 6.8e−3 | 0.0034 | 5.2e−4 | 0.019 | 0.0023 | |

**Table 2 One-sided signed-rank significance values for the probability stem pcs with high vs. low trajectory reliability stay active between sessions ($n$ = session pairs).**

| Day lag (n) | 1 (57) | 2 (34) | 3 (24) | 4 (24) | 5 (29) | 6 (41) | 7 (44) | 8 (32) |
|---|---|---|---|---|---|---|---|---|
| All Neurons | 5.4e−7 | 6.0e−6 | 0.0027 | 2.1e−4 | 0.0017 | 0.0011 | 6.5e−4 | 0.010 |
| Event-rate matched | 0.24 | 0.18 | 0.38 | 0.30 | 0.66 | 0.33 | 0.30 | 0.41 |
| Day lag (n) | 9 (22) | 10 (10) | 11 (10) | 12 (14) | 13 (22) | 14 (17) | 15 (14) | |
| All Neurons | 0.013 | 0.29 | 0.082 | 0.029 | 0.0020 | 0.0026 | 0.066 | |
| Event-rate Matched | 0.34 | 0.58 | 0.31 | 0.98 | 0.83 | 0.26 | 0.5 | |

High/low reliability = stem PCs in top/bottom quartile of mean reliability value.

**Table 3 One-sided signed-rank significance values for the probability splitter vs. place and non-place cells stay active, event-rate matched ($n$ = session pairs).**

| Day lag (n) | 1 (58) | 2 (34) | 3 (24) | 4 (25) | 5 (30) | 6 (41) | 7 (44) | 8 (31) |
|---|---|---|---|---|---|---|---|---|
| vs. Arm PCs | 0.56 | 0.12 | 0.98 | 0.98 | 0.86 | 0.20 | 0.31 | 0.33 |
| vs. Stem PCs | 0.47 | 0.68 | 0.95 | 0.68 | 0.82 | 0.40 | 0.97 | 0.95 |
| vs. Arm NPCs | 7.9e−7 | 0.0011 | 0.0083 | 0.065 | 0.073 | 0.014 | 0.0022 | 0.0090 |
| vs. Stem NPCs | 0.0040 | 0.0030 | 0.024 | 0.37 | 0.25 | 0.049 | 0.0010 | 0.062 |
| Day lag (n) | 9 (21) | 10 (10) | 11 (10) | 12 (14) | 13 (22) | 14 (17) | 15 (13) | |
| vs. Arm PCs | 0.13 | 0.35 | 0.65 | 0.57 | 0.26 | 0.20 | 0.07 | |
| vs. Stem PCs | 0.83 | 0.82 | 0.92 | 0.93 | 0.90 | 0.65 | 0.71 | |
| vs. Arm NPCs | 0.016 | 0.60 | 0.27 | 0.027 | 0.46 | 0.23 | 0.0017 | |
| vs. Stem NPCs | 0.034 | 0.63 | 0.71 | 0.27 | 0.11 | 0.27 | 0.034 | |

Neurons with the lowest event-rate were removed from each session such that the mean event rate of each coding type matched that of splitter neurons.

place cells also have larger place fields, but not larger spatial correlations, than arm place cells (Supplementary Fig. 5a, b). This indicates that trajectory-dependent splitter neurons might guide memory task performance by providing a more consistent representation of space than place cells.

**Rapid onset of trajectory coding followed by stable activity.** Next, we examined the ontogeny of trajectory-dependent neural behavior. We hypothesized two different scenarios that could support the emergence of splitters. In line with a study showing that unstable neurons can support well-learned behavior[36], splitters could slowly ramp up/down their splitting behavior or they could come online suddenly and turn off just as suddenly. On the other hand, previous research presented the idea that neurons pre-disposed to become place cells can come online suddenly after a head-scanning/attention event[37], which is potentially supported by the presence of reliable sub-threshold depolarizations of those neurons caused by calcium activity in its dendritic arbor[38–40]. In line with this idea, splitter neurons could rapidly develop trajectory-dependent activity and then maintain

that activity thereafter. To address this question, we identified the day when each neuron we recorded first exhibited significant trajectory-dependent activity and tracked its splitting extent—the proportion of the stem which exhibited significant differences between left and right tuning curves—in subsequent sessions. We found evidence for heterogeneity in the onset of splitting, with some neurons exhibiting a rapid onset of trajectory-dependent activity (Fig. 6a) while others ramped up their trajectory-dependent activity in the days prior to becoming a splitter (Fig. 6b). Each onset type appeared to maintain stable trajectory-dependent activity afterward since, for individual mice, splitting extent remained higher in the day following splitter onset when compared to the day preceding splitter onset (Fig. 6c, Supplementary Fig. 2d). The rapid onset of trajectory-dependent activity and stable maintenance thereafter was readily apparent when examining group data over longer timescales (±10 days, Fig. 6d, Supplementary Fig. 2d–n). We obtained similar results for peak reliability and peak discriminability along the stem (Fig. 6e, f). In contrast, we observed a weaker trend for discriminability and no trend for reliability averaged along the whole stem (Supplementary Fig. 6), supporting

the observation that splitter neurons maintained consistent trajectory-dependent activity along a local portion (~25%) of the stem after their onset. We observed a similar onset profile for place cells when we used mutual information as a metric of place coding strength (Fig. 6g), which suggests that similar rules govern the onset and fate of trajectory-dependent and spatial coding in hippocampal neurons.

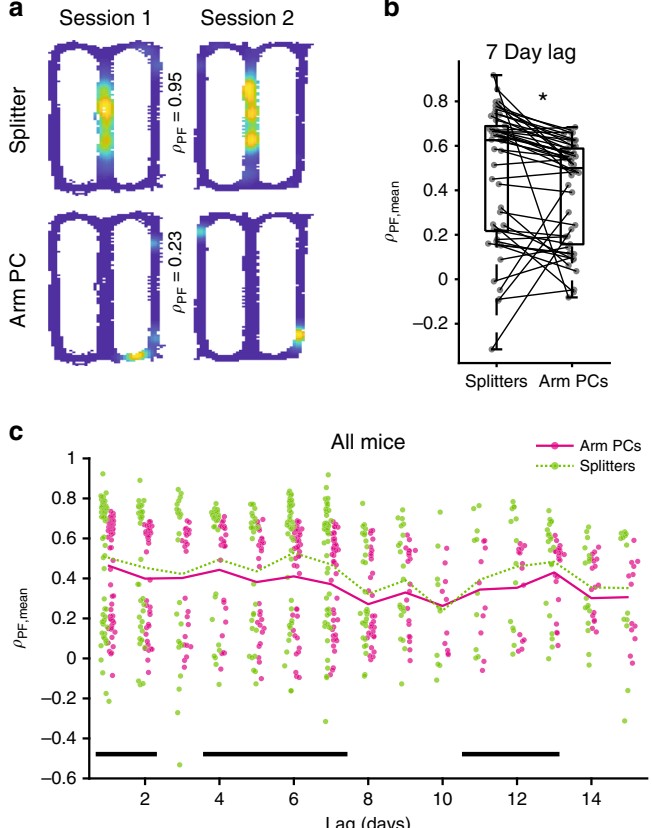

**Fig. 5 Splitters maintain more consistent spatial information than place cells. a** Example 2D occupancy normalized calcium event maps from the same splitter neuron (top) and arm place cell (bottom) between sessions on adjacent days. The higher Spearman correlation between splitter neuron event maps indicates more consistent spatial activity. Event maps were smoothed with a Gaussian filter (σ = 2.5 cm). **b** Mean spatial correlations from smoothed event maps for splitter neurons versus return arm PCs for all sessions seven days apart from all. *$p = 9.7 \times 10^{-5}$, $n = 42$ session pairs, one-sided signed-rank test. Box plots show median and 1st/3rd quartiles, whiskers extend to 1.5× interquartile range. **c** Mean spatial correlations for splitter neurons and arm PCs versus lag between sessions for all mice/ sessions. Magenta = arm PCs, green dashed = splitters. Black bars = significant differences after Holm-Bonferroni correction (15 day lags considered) of one-sided signed-rank test, α = 0.05. See Table 4 for raw p-values at all lags.

**Place cell onset coincides with or precedes splitter onset**. We next wondered if hippocampal neurons displayed significant spatial tuning before, during, or after they exhibited trajectory-dependent firing. As shown above, splitter cells produce accurate spatially modulated activity (Fig. 5) and have a similar onset/ offset trajectory to place cells (Fig. 6); thus, we hypothesized that the onset of trajectory-dependent firing in hippocampal neurons would either coincide with or follow their onset as place cells. To test this idea, we first tallied the onset day of splitter neurons and place cells. We found that, while both place cells and splitter neurons were present from day 1 and continued to come online throughout the experiment (Fig. 7a, b), the bulk of place cells were recruited on day 1. In contrast, and in agreement with a previous study[41], the recruitment of splitter cells did not peak until several days later (Fig. 7a, b), suggesting that trajectory-dependent activity tended to emerge more slowly than spatial activity. This could occur independently in two different groups of neurons, or it could occur serially with each neuron first becoming a splitter cell only after becoming a place cell. Thus, to test if this delay in splitter cell ontogeny occurred in the same cells, we directly compared the day a cell became a place cell to the day it began to exhibit trajectory-dependent activity. We found that in the majority of neurons, trajectory-dependent activity onset occurred simultaneously with place field onset, while a different population of neurons exhibited trajectory-dependent activity only after first becoming place cells (Fig. 7c, d). Consistent with previous studies[37,42] the bulk of splitters and place cells came online in the first several trials of each session; surprisingly, splitter activity first occurred earlier than place cell activity over trials within a day (Fig. 7e, f, Supplementary Fig. 7) even though they appeared later across days (Fig. 7a, b). Thus, place cells and splitter cells occupy an overlapping population of neurons with spatial responsivity coinciding with or preceding trajectory-dependent coding.

## Discussion

From an evolutionary perspective, one adaptive function of memory is the ability to provide information vital to survival. Thus, maintaining activity and consistency in neurons encoding information pertinent to survival might provide a mechanism for preferentially strengthening connections with downstream structures via consistent replay of the same sequences[43–46]. Conversely, if the pool of neurons available to encode a given memory remains fixed, then forgetting of incidental information through the turnover/silencing of neurons not required for survival is adaptive[47] because it could increase the numbers of neurons available to encode other relevant information[48]. Here, we utilized in vivo calcium imaging with miniaturized microscopes to explore this idea (Fig. 1) by investigating the development and fate of trajectory-dependent splitter neurons[15,16] (Fig. 2). To the best of our knowledge, this is the first demonstration that trajectory-dependent hippocampal activity exists in mice and that it can be detected with calcium imaging. Since

**Table 4 One-sided signed-rank significance values for mean spatial correlation values of splitter vs. arm PCs or splitters vs. stem PCs ($n$ = session pairs).**

| Day lag (n) | 1 (61) | 2 (38) | 3 (26) | 4 (26) | 5 (30) | 6 (42) | 7 (46) | 8 (31) |
|---|---|---|---|---|---|---|---|---|
| vs. Arm PCs | 0.0055 | 0.013 | 0.049 | 0.014 | 0.0057 | 9.5e−6 | 9.7e−5 | 0.018 |
| vs. Stem PCs | 1.9e−5 | 2.8e−5 | 0.014 | 2.5e−4 | 8.2e−4 | 3.5e−5 | 1.0e−4 | 0.017 |
| Day lag (n) | 9 (21) | 10 (10) | 11 (10) | 12 (14) | 13 (22) | 14 (17) | 15 (13) | |
| vs. Arm PCs | 0.044 | 0.65 | 0.0098 | 8.5e−4 | 0.0081 | 0.61 | 0.047 | |
| vs. Stem PCs | 0.028 | 0.19 | 0.0049 | 0.0083 | 0.011 | 0.56 | 0.04 | |

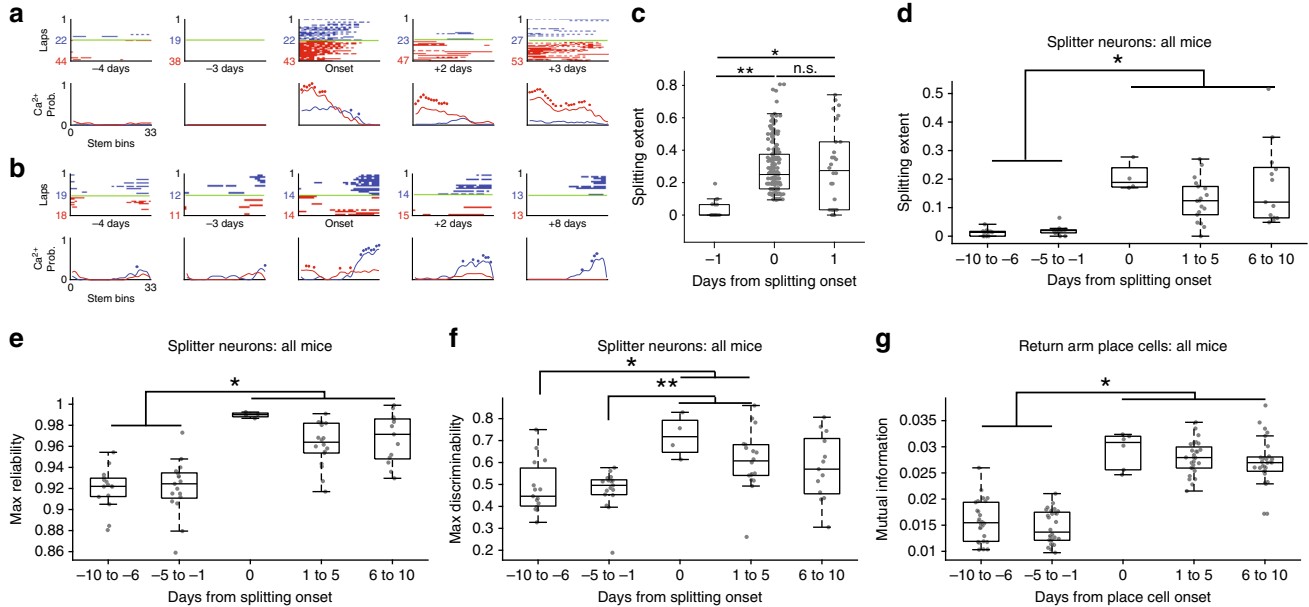

**Fig. 6 Splitters come online abruptly and maintain stable fields. a, b** Example splitter across days for two different mice illustrating sudden onset of trajectory-dependent activity followed by stable trajectory-dependent activity thereafter. **c** Splitting extent (proportion of stem bins with significant trajectory-dependent activity) along the stem±1 days from splitter onset for one representative mouse. $p = 1.8 \times 10^{-10}$ Kruskal–Wallis ANOVA, $N = 233$ neurons, *$p = 1.2 \times 10^{-5}$, **$p = 1.0 \times 10^{-9}$ two-sided post-hoc Tukey test. Gray circles = splitting extent for each neuron. Box plots show median and 1st/3rd quartiles, whiskers extend to 1.5× interquartile range. **d** Extent of splitting±10 days from splitter onset for all mice. $p = 2.5 \times 10^{-9}$ Kruskal–Wallis ANOVA, $N = 67$ means, *$p < 6.2 \times 10^{-3}$ two-sided post-hoc Tukey test. Gray circles = mean splitting extent for each mouse of all neurons active on the stem across all session-pairs. Box plots show median and 1st/3rd quartiles, whiskers extend to 1.5x interquartile range, same conventions used in (**e–g**). **e** Max reliability score±10 days from splitter onset for all mice. $p = 4.4 \times 10^{-8}$ Kruskal–Wallis ANOVA, $N = 67$ means, *$p < 1.6 \times 10^{-3}$ two-sided post-hoc Tukey test. **f** Peak discriminability score±10 days from splitter onset for all mice. $p = 3.4 \times 10^{-4}$ Kruskal–Wallis ANOVA, $N = 67$ means, *$p < 0.016$, **$p < 0.018$ two-sided post-hoc Tukey test. **g** Mean mutual information±10 days from return place cell onset for all mice. $p = 3.0 \times 10^{-11}$ Kruskal–Wallis ANOVA, $N = 67$ means, *$p < 3.7 \times 10^{-3}$ two-sided post-hoc Tukey test.

trajectory-dependent splitter neurons contain information relevant to proper task performance (Fig. 3, see also Ferbinteanu and Shapiro[25]), we hypothesized that they would exhibit relatively high stability when compared to other neuron functional coding types.

Several lines of evidence support this hypothesis. First, splitter neurons are more likely to remain active across long timescales than arm place cells and especially non-place cells (Fig. 4). Second, splitters come online abruptly and then maintain a stable readout of trajectory up to 10 days after becoming a splitter (Fig. 6). Splitters also provide a more consistent signal of the animal's current location than do other neurons (Fig. 5), further supporting their long-term stability. Last, we found that splitter cells are a dynamic subpopulation of place cells with the onset of place coding generally preceding the onset of trajectory-dependent activity (Fig. 7). This finding concurs with the slow increase of trajectory-dependent activity with experience found in a previous study[41]. These data combined support the idea the information carried in the neural code influences a neuron's stability[19] and the consistency of the information it provides to downstream structures. More broadly, this study supports the idea that adaptive memories are encoded in a relatively stable subpopulation of neurons, freeing the remaining pool of neurons to undergo plasticity during new learning[49,50]. Interestingly, since information transfer to the lateral septum correlates more strongly with the strength of hippocampal activity rather than information content[51], the finding that event-rate also influences the stability of a neuron across days (Fig. 4 vs. Supplementary Fig. 4) could provide a mechanism for maintaining stable outputs to a prominent subcortical output of the hippocampus. However, how downstream regions can utilize a constantly changing landscape of hippocampal inputs to guide

behavior remains an open question, as place fields along the stem drift steadily backwards throughout each session (Supplementary Fig. 2a, b) and day-to-day turnover even in relatively stable splitter neurons can still sometimes be quite high (Fig. 4c–f).

Our study utilizes single-photon imaging to perform longitudinal tracking of hippocampal neuron activity and confirms existing studies that show increasing turnover of coactive neurons with time[21–23]. However, a recent study by Katlowitz et al. performed in songbirds demonstrated that imaging artifacts, specifically small shifts in the z-plane of single-photon imaging, could entirely account for putative cell turnover[52]. Thus, the turnover we and others observe in hippocampal neurons could likewise be artefactual. While relevant, this concern is mitigated in our study for a number of reasons. First, the Katlowitz et al. study[52] was performed in the basal ganglia of songbirds while they performed a stereotyped behavior supported by highly stable firing responses of neurons over short and long timescales[53,54]. In contrast, our study was performed in CA1 of the mouse hippocampus, a highly plastic brain region exhibiting complete, monthly turnover of afferent connections[55] that also exhibits a high degree of drift in neuron firing responses over relatively short timescales[13,56]. Second, studies utilizing activity-dependent tagging of neurons also find that the overlap between active cells in the mouse hippocampus declines with time between sessions[23,57], supporting long-term hippocampal cell turnover as a real phenomenon. Third, a notable recent study used two-photon imaging, which mitigates any concerns of z-plane drift, and found similar rates of turnover in CA1 to what we observe[58]. Most importantly, our study compares the relative turnover rates of two different classes of cells from the same session: splitter cells and place cells. Thus, even if day-to-day misalignments in the z-plane forced neurons

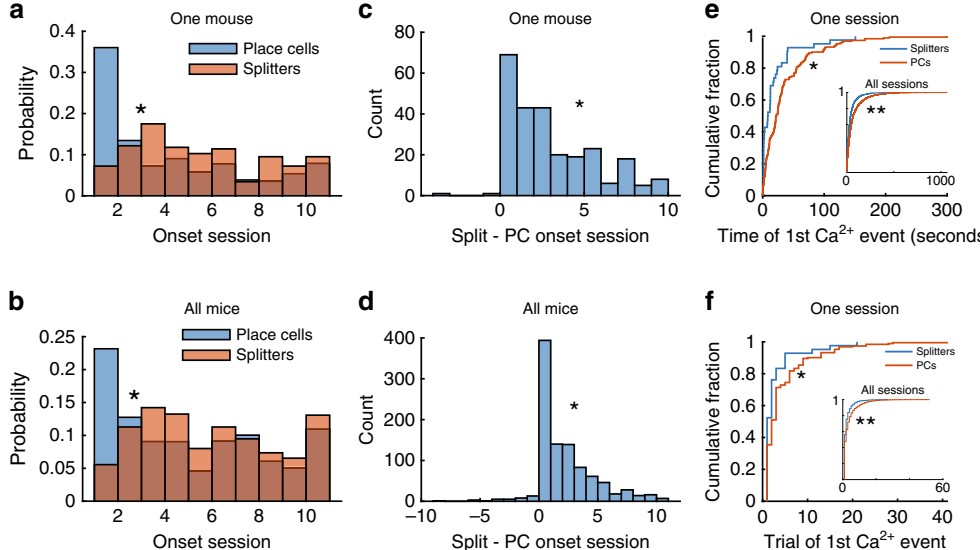

**Fig. 7 Place cell recruitment peaks prior to splitter onset between but not within sessions. a** Histogram of the first (onset) day a neuron is identified as a splitter cell or place cell for one mouse. *$p = 8.2 \times 10^{-19}$, one-sided Kolmogorov–Smirnov test, $n = 1628$ place cells and 263 splitters. **b** Same as (**a**) but for all mice. *$p = 2.0 \times 10^{-21}$, one-sided Kolmogorov–Smirnov test, $n = 5062$ place cells and 612 splitters. **c** Difference between splitter cell onset day and place cell onset day for one mouse. *$p = 8.2 \times 10^{-18}$ $\chi^2$ goodness-of-fit test, mean = 2.54, median = 2, $n = 256$ neurons. **d** Same as (**c**) but for all mice. *$p = 5.5 \times 10^{-79}$ $\chi^2$ goodness-of-fit test, mean = 1.7, median = 1, $n = 982$ neurons. **e** Cumulative probability plot of splitter vs. place cell recruitment time for one mouse. (inset) all mice/sessions. *$p = 3.5 \times 10^{-4}$ ($n = 42$ splitters and 192 place cells), **$p = 3.9 \times 10^{-50}$ ($n = 1190$ splitters and 11232 place cells), one-sided Kolmogorov–Smirnov test. **f** Cumulative probability plot of splitter vs. place cell recruitment trial for same session shown in (**e**) (inset) all mice/sessions. *$p = 0.01$ ($n = 42$ splitters and 192 place cells), **$p = 1.2 \times 10^{-24}$ ($n = 1190$ splitters and 11232 place cells), one-sided Kolmogorov–Smirnov test.

out of focus, this would occur equally for splitters, place cells, and non-place cells. Therefore, concerns about imaging artifacts cannot explain our finding that splitter cells are more persistently active across long timescales than arm place cells and all non-place cells.

Rodents with hippocampal lesions are capable of performing a continuous alternation task[59]. This raises the question: how important is trajectory-dependent activity if mice can perform the task without the hippocampus at all? We have several responses. First, long-term lesions test necessity, not sufficiency, since these lesions can induce compensatory plasticity that could allow non-hippocampal regions to support the task[60]. Second, under normal conditions the hippocampus might still be the default brain region for task performance in spatial alternation. This is emphasized by Goshen et al.[61], who demonstrated that mice cannot perform long-term recall of a putatively hippocampal-independent contextual fear memory[57,62] when hippocampal inactivation is limited to a short time period before the task; however, mice became capable of successful long-term memory recall when this inactivation was extended over a long time period prior to performing the task. This study and others[63–67] support the idea that the hippocampus is vital for long-term recall under normal conditions and that redundant pathways are recruited for episodic memory retrieval only if chronic aberrant activity is detected in the hippocampus. Finally, the results of Ferbinteanu & Shapiro[25] demonstrate that decreases in CA1 trajectory-coding correlate with impaired performance in a task with discrete trial structure. This suggests that the trajectory-dependent activity observed during hippocampal-independent tasks becomes necessary for proper memory-retrieval when task demands require higher levels of hippocampal engagement.

One notable study by Ito et al. found that lesions or optogenetic silencing of nucleus reuniens, an important communication hub between the medial prefrontal cortex and dorsal CA1 of the hippocampus, significantly reduced trajectory-dependent activity in rat CA1 neurons while having no impact on a rat's performance of a spatial alternation task[31]. Those results challenge the findings of us and others[25,27] that, in some cases, the quality of trajectory-dependent information contained in CA1 activity patterns correlates with a mouse's performance (Fig. 3). One potential reason for this discrepancy is that their intervention only partially reduced trajectory-dependent information without eliminating it, allowing the splitter cells remaining to provide adequate information for proper task performance. In fact, optogenetic silencing of nucleus reuniens produced a smaller deficit in trajectory-dependent activity than did lesions; even lesions eliminated trajectory-dependent activity predicting future trajectories only. Information related to past trajectories, which could be utilized by downstream structures to help make the correct upcoming turn, was maintained. Second, relatively easy tasks might be less resistant to a partial disruption and rats performed at close to ceiling levels in the Ito et al.[31] study. Our mice performed at lower levels, though still well above chance, indicating that the spatial alternation task might place higher attentional and cognitive demands on mice than on rats. Finally, other recent studies utilizing hippocampal-dependent memory tasks have shown that nucleus reuniens disruption causes memory deficits[68], while silencing of medial prefrontal cortex reduces both behavioral performance and the magnitude of trajectory-dependent activity in CA1[69]. These studies highlight the importance of trajectory-dependent activity to the performance of more difficult, hippocampal-dependent memory tasks. Taken together, our findings, along with the work of others[25,27] indicate that further work is necessary to disentangle the exact conditions under which trajectory-dependent activity is necessary or sufficient for performing a spatial alternation task.

Through what mechanism do trajectory-dependent neurons maintain greater stability across long timescales? After the initial onset of trajectory-dependent behavior, these neurons could receive feedback from dopaminergic neurons originating in the ventral tegmental area (VTA) during learning[70] or from locus coeruleus (LC) neurons during post-learning sleep[71] that could

strengthen afferent connections to splitter neurons. This could also occur during sharp-wave ripple related replay of prior trajectories[43,44] in conjunction with simultaneous dopaminergic inputs from VTA[70]. However, this mechanism would also strengthen all cells active en route to the goal location whether they carried information about trajectory or not. One recent study found that trajectories leading to larger rewards were preferentially replayed over trajectories leading to smaller rewards[72]. Thus, one possibility is that since trajectory-dependent neurons are more useful for predicting how to obtain reward than pure place cells, they might be preferentially reactivated during sharp-wave ripple events, an idea that warrants future testing.

Taken together, our results highlight the influence of a neuron's information content on its subsequent stability, and suggest that the emergence of task-related trajectory-dependent coding coincides with or follows the emergence of spatial coding in neurons. Future work should investigate mechanisms supporting the stability and emergence of trajectory-dependent neurons.

## Methods

**Animals**. Five male C57/BL6 mice (Jackson Laboratories), age 3–14 months and weighing 25–30 g were used. One mouse was excluded from analysis after performing the experiment due to the inability to correct motion artifacts in his imaging videos. Mice were housed socially with 1–3 other mice in a vivarium on a 12 h light/dark cycle with lights on at 7 am and given free access to food and water. All mice were singly housed after surgery. All procedures were performed in compliance with the guidelines of the Boston University Animal Care and Use Committee, including a mandated housing temperature of 18–25 °C and 30–70% humidity.

**Viral constructs**. We used an AAV9.*Syn*.GCaMP6f.WPRE.SV40 virus from the University of Pennsylvania Vector Core at an initial titer of ~4 × 10^13 GC/mL and diluted it to ~(5–6) × 10^12 GC/mL with sterilized 0.05 phosphate buffered saline (KPBS) prior to infusion into CA1.

**Stereotactic surgeries**. All surgeries were performed in accordance with previously published procedures[30] in accordance with the Boston University Animal Care and Use Committee. Briefly, we performed two stereotactic surgeries and one base-plate implant on naïve mice, aged 3–8 months. Surgeries were performed under 1–2% isoflurane mixed with oxygen. Mice were given 0.05 mL/kg buprenorphine (Buprenex) for analgesia, 5.0 mL/kg of the anti-inflammatory drug Rimadyl (Pfizer), and 400 mL/kg of the antibiotic Cefazolin (Pfizer) immediately after induction. They received the same dosage of Buprenex, Cefazolin, and Rimadyl twice daily for three days following surgery and were carefully monitored to ensure they never dropped below 80% of their pre-operative weight during convalescence. In the first surgery, a small craniotomy was performed at AP −2.0, ML +1.5 (right) and 250nL of GCaMP6f virus was injected 1.5 mm below the brain surface at 40nL/min. The needle remained in place a minimum of 10 min after the infusion finished at which point it was slowly removed, the mouse's scalp was sutured, and the mouse was removed from anesthesia and allowed to recover.

Three to four weeks after viral infusion, mice received a second surgery to attach a gradient index (GRIN) lens (GRINtech, 1 mm × 4 mm). After performing an ~2 mm craniotomy around the implant area, we carefully aspirated cortex using a blunted 25ga and 27ga needle under constant irrigation with cold, sterile saline until we visually identified the medial-lateral striations of the corpus callosum. We carefully removed these striations using a blunted 31ga needle while leaving the underlying anterior-posterior striations intact, after which we applied gelfoam to stop any bleeding. We then lowered the GRIN lens until it touched the brain surface and then proceeded to lower it another 50 μm to counteract brain swelling during surgery (note that in two mice we first implanted a sleeve cannula with a round glass window on the bottom without depressing an additional 50 μm and then cemented in the GRIN lens during base-plate attachment). We then applied Kwik-Sil (World Precision Instruments) to provide a seal between skull and GRIN lens and then cemented the GRIN lens in place with Metabond (Parkell), covered it in a layer of Kwik-Cast (World Precision Instruments), and then removed the animal from anesthesia and allowed him to recover after removing any sharp edges remaining from dried Metabond and providing any necessary sutures.

Finally, after ~2 weeks we performed a procedure in which the mouse was put under anesthesia but no tissue was cut in order to attach a base-plate for easy future attachment of the microscope. To do so, we attached the base-plate to the camera via a set screw, carefully lowered the camera objective and aligned it to the GRIN lens by eye, and visualized fluorescence via nVistaHD v2.0/v3.0 until we observed clear vasculature and putative cell bodies expressing GCaMP6f, then raised the camera up ~50 μm before applying Flow-It ALC Flowable Composite (Pentron) between the underside of the baseplate and the cured Metabond on the mouse's

skull. After light curing we applied opaque Metabond over the Flow-It ALC epoxy to the sides of the baseplate to provide additional strength and to block ambient light infiltration.

All mice were imaged in open arenas for several weeks to habituate them to attaching and wearing the camera (see following section). Additionally, two of the mice participated in another experiment prior to imaging. As a result, the mean time the virus was in the system on the last imaging day was 19.4 weeks.

**Experimental outline**. After recovery from surgery, mice were food deprived to maintain no less than 85% of their pre-surgery weight. Mice were subsequently exposed to a variety of arenas in order to habituate them to navigating with the camera attached. Prior to training on the alternation task, all mice were given 1–4 habituation sessions on the alternation maze. The maze floor (inner dimension = 64 × 29 cm) and walls (height = 18 cm) were constructed from 3/8 inch (0.95 cm) thick plywood and the barriers between arms were constructed from two 53 cm long 1.5 × 5.5 inch (3.8 × 14 cm) pine framing studs. The finished maze consisted of a central stem and two return arms, each 7.5 cm wide with 5.7 cm wide openings at each end of the central stem through which mice could exit or enter the return arms. Two food wells ~0.25 cm deep were created toward the end of each return arm to hold chocolate sprinkles: they were centered 12.5 cm from the end of the maze where mice exited the return arm/entered the center stem. Food was placed in these wells through a small opening in the side of the maze. The arena was sealed with urethane prior to exploration.

Three of the mice were first trained to loop on each side of the maze independently for 3 days in 10 min blocks by blocking off access to the other side with Plexiglas dividers in order to familiarize mice with the general task demands, arena, and location of food reward (chocolate sprinkles); the other mouse received one habituation session where he was allowed to freely traverse the maze. Following habituation, mice were placed in the center stem and rewarded at the well on the reward arm regardless of the first turn direction. On subsequent trials, mice were only rewarded if they turned the opposite direction of the previous trial. Mice were allowed to run freely and were only blocked when they (a) attempted to reverse course on the central stem, (b) attempted to exit the return arm after they had committed to it, or (c) attempted to run straight across to the other arm without returning to the central stem after obtaining reward. A mouse was considered committed to an arm after his tail entirely crossed from the edge of the central arm into the stem. Mice generally ran ballistically up the center stem and were allowed to pause once they entered the return arm and after they obtained reward. Food reward was only delivered once the mouse had committed to a return arm in order to avoid providing an auditory cue of reward location. Two mice were forced to alternate in a subset of sessions/trials. One mouse encountered a lapse in performance mid-way through the experiment and began perseverating on one turn direction in blocks: he was subsequently given a number of trials at the beginning of each session where he was forced to turn each direction by blocking off one turn direction with a Plexiglas divider, after which he was then allowed to freely choose turn directions. Several sessions in which this mouse failed to consistently run ballistically down the center stem were excluded from analysis. The other mouse was initially forced to alternate at the end of his habituation looping sessions. All forced trials were not considered during later data analysis. Mice performed 1–2 sessions per day, and one mouse received a break of ~15 min after the first set of 20 trials in a subset of sessions. Sessions were terminated each day after 30 min or when the mouse stopped consistently running ballistically down the center arm, whichever came first. The experiment lasted 27, 16, 29, and 36 days for the four mice involved.

**Imagine acquisition and processing**. Brain imaging data was obtained using nVista HD (Inscopix) v2/v3 at 1440 × 1280 pixels and a 20 Hz sample rate. Two mice were lightly anesthetized (~60 s) to facilitate camera attachment and then given ~15 min to recover prior to any recordings; the camera was attached to the other two mice while they were awake. Prior to neuron/calcium event identification we first pre-processed each movie using Mosaic (Inscopix) software which entailed a) spatially downsampling by a factor of 2 (1.18 μm/pixel), b) performing motion corrections, and c) cropping the motion-corrected movie to eliminate any dead pixels or areas with no calcium activity. We then extracted a minimum projection of the pre-processed movie for later neuron registration. We replaced isolated dropped frames (maximum two consecutive frames) with the previous good frame, and in the rare case where more than two frames dropped in a row these frames were excluded from all analyses. For one mouse, we observed poor imaging quality on 3 of 10 sessions. The minimum projection from each of these sessions were significantly different from the other seven sessions and from each other, indicating improper camera alignment. These sessions were excluded from analysis, resulting in n = 68 viable recording sessions (n = 10, 7, 23, and 28 for mice 1–4).

**Neuron and calcium event identification**. We utilized custom-written, open-source MATLAB software (available at https://github.com/SharpWave/Tenaspis) to identify putative neuron ROIs and their calcium events in accordance with previously published results[24,30]. A neuron had to have at least four calcium events in order to be considered active on a given session.

We calculated baseline fluorescence for each neuron as the mean pixel intensity, derived from the minimum projection of a given session, across all pixels in that neuron's ROI.

**Neuron exclusion criteria**. In line with Tian et al.[73], we calculated calcium event decay times in the following manner. First, we fit an exponential function to the decaying portion of the last recorded transient using MATLAB's fit function with the "exp1" parameter and calculated that function's half-life. In the case that the frames following that trace's end were invaded by fluorescence from a neighboring neuron, resulting in an exceedingly long decay time (>7 s), we iteratively utilized the previous trace. We were unable to find a transient isolated from neighboring neuron fluorescence in <0.5% of neurons—these were excluded from further analysis. We excluded any neurons with half-decay times >2 s (calculated using the second method, which generally produced longer half-decay times and was thus more conservative) from further analysis.

The small number of neurons that were modulated by lateral position rather than trajectory were also excluded from analysis, see "Trajectory-Dependent/Splitter Cell Identification" section (below) for details.

**Across-session neuron registration**. We utilized custom-written, freely available MATLAB code (available at https://github.com/nkinsky/ImageCamp) to perform neuron registration across sessions in accordance with previously published results (see Supplementary Fig. 1). We checked the quality of neuron registration between each session-pair in two ways: (1) by plotting the distribution of changes in ROI orientations between session and comparing it to chance, calculated by shuffling neuron identity between session 1000 times, and (2) plotting ROIs of all neurons between two sessions and looking for systematic shifts in neuron ROIs that could lead to false negatives/positives in the registration. During the course of these checks, we noticed the quality of registration between sessions dropped significantly approximately halfway through the experiment for two mice (mouse 3 and mouse 4). Thus, we excluded any registrations occurring between the first and second halves of the experiment for these two mice. Furthermore, the second half of the experiment was excluded for these two mice when calculating the absolute onset session of place cells versus splitter cells (Fig. 7a, b) but was included when calculating the relative onset day for each cell type (Fig. 7c, d). Several other session pairs exhibiting poor registrations based on the criteria above were also excluded, though these were rare.

**Behavioral tracking and parsing**. Behavioral data were recorded via an overhead camera with Cineplex v2/v3 software (Plexon) at a 30 Hz sample rate. Cineplex produced automated tracking of the animal's position by comparing each frame to a baseline image without the animal in the arena. Imaging and behavioral data were synchronized by TTL pulse at the beginning of the recording. Each video was inspected by eye for errors in automated tracking and fixed manually via custom-written MATLAB software. After fixing all erroneous data points, the animal's position was interpolated to determine its location at each imaging movie time point.

**Histology**. Mice were killed and transcardially perfused with 10% KPBS followed by formalin. Brains of perfused mice were then extracted and post-fixed in formalin for 2–4 more days after which they were placed in a 30% sucrose solution in KPBS for 1–2 additional days. The brains were then frozen and sliced on a cryostat (Leica CM 3050 S) in 40 μm sections after which they were mounted and coverslipped with Vectashield Hardset mounting medium with DAPI (Vector Laboratories). We then imaged slides at 4×, 10×, and 20× on a Nikon Eclipse Ni-E epifluorescence microscope to verify proper placement of the GRIN lens above the CA1 pyramidal cell layer.

**Place cell identification**. We utilized identical methods to those outlined in Kinsky et al.[30] to identify place cells, reproduced here:

"Calcium transients were spatially binned (4 cm × 4 cm) and normalized by occupancy. Spatial mutual information (SI) was computed from the following equations, adapted from Olypher et al.[74]:

$$I_{pos}(x_i) = \sum_{k=0}^{1} P_{k|x_i} \log\left(\frac{P_{k|x_i}}{P_k}\right),$$ (1)

$$SI = \sum_{i=1} P_{x_i} I_{pos}(x_i),$$ (2)

where $P_{xi}$ is the probability the mouse is in pixel $x_i$, $P_k$ is the probability of observing $k$ calcium events (0 or 1), $P_{k|xi}$ is the conditional probability of observing $k$ calcium events in pixel $x_i$.

The SI was then calculated 1000 times using shuffled calcium event timestamps, and a neuron was classified as a place cell if it (1) had at least five calcium transients during the session, and (2) the neuron's SI exceeded 95% of the shuffled SIs. We obtained similar results using smoothed occupancy rate maps, which were constructed using 1 cm × 1 cm bins and applying a Gaussian filter ($\sigma = 2.5$ cm). We defined the extent of a place field as all connected occupancy bins whose smoothed event rate exceeded 50% of the peak event rate occupancy bin." Note that only time bins in which the mouse was moving faster than 1 cm/s were included.

Place-field correlations between sessions were calculated using the smoothed occupancy normalized rate maps. Finally, to determine place field length, the place field area (the number of contiguous bins above the 50% peak rate threshold) was divided by 5 cm (the effective width of each corridor occupied by the mice after accounting for occupancy and any angular distortions in video tracking).

**Trajectory-dependent or splitter cell identification**. Prior to performing any analysis, each mouse's trajectory data was aligned to that from the first habituation session. This was done by (1) manually rotating the data to correct for any day-to-day changes in maze angle relative to the recording camera, (2) calculating the edges of the mouse's trajectory as the data points located at the 2.5% and 97.5% points in the cumulative density function of his $x/y$ position data, and (3) adjusting the data by applying the necessary translation and scaling (minimal) to overlay each session's trajectory on the first session. After aligning data across sessions, the mouse's trajectory on each trajectory was parsed into his progression through the different sections of the maze, starting at the (a) *base*, then moving down the (b) *center stem* into the (c) *choice* point, then turning into the (d) left/right *entry* to the (e) *return arm*, and finally entered the (f) *approach* to the center stem just after the reward port. The center stem portion was manually identified for each mouse as the point where the mouse's trajectory into/out of each return arm stopped diverging. This was done in order to mitigate the possibility that trajectory-depending activity was controlled entirely by stereotyped sensory inputs, e.g. the mouse hugging/whisking the left side of the center stem after right turn trial.

After parsing the animal's behavior into these sections, the center stem was broken up into ~1 cm bins and the event rate for each neuron was calculated for each trial. Occupancy normalized tuning curves for each trial type (left or right turn) were then constructed, which consisted of each neuron's mean event rate for all correct trials at each spatial bin divided by the time the mouse spent in each bin. The difference between these curves was then calculated. Tuning curves were smoothed using the fit function in MATLAB with $p = 0.9$ for visualization purposes (Fig. 2). To assess significance, we again constructed un-smoothed tuning curves for left/right trials and calculated their difference, but after randomly shuffling trial turn identity 1000 times to establish the likelihood the observed difference between tuning curves could emerge by chance. We then defined splitters/trajectory-depending cells as neurons which had at least three bins whose real tuning curve difference exceeded 950 of the 1000 shuffled values. In order to exclude spurious identification of splitters we only included neurons that produced a calcium event on the stem of the maze on at least five trials.

We calculated several different metrics to quantify the level of trajectory-dependent activity in each neuron. First, we calculated discriminability by summing the absolute value of the difference between tuning curves along all stem bins and then dividing by the sum of tuning curves along all stem bins. Second, we calculated reliability in the following manner: (a) we shuffled trial identity 1000 times and calculated the difference between shuffled tuning curves, then (b) calculated the proportion of shuffles in which the real difference between tuning curves exceeded that of shuffled, then (c) calculated reliability as the mean of this proportion along all the stem bins. Note that splitter neurons by definition must have at least three bins with a reliability value above 0.95 (see above). Last, we calculated the correlation between left and right unsmoothed tuning curves (~1 cm bins). Note that this metric is very conservative since it produces low correlations for splitters who shift the location of their peak activity between left and right trials along the length of the stem (Fig. 2b) but not for splitters who modulate their event rate in the same place along the stem (Fig. 2a). Finally, we defined splitting extent for each neuron as the proportion of stem bins that exhibited significant differences between left and right tuning curves.

In order to check the robustness of our results and control for any trajectory-dependent information resulting from stereotyped deviations in speed or lateral position along the stem, we also performed an additional analysis in line with previous studies[16,31]. To do so, we first divided the stem lengthwise into five bins and calculated the average transient probability in each bin for all trials. Note that we did not segment the stem laterally but instead used lateral position as a continuous predictor (as described in the following sentences) in line with Wood et al.[16]. We then performed an ANOVA analysis using the anovan function in MATLAB for each trajectory-dependent splitter neuron we detected. We used trial type (left/right), stem bin, stem bin x trial type as categorical predictors, the animal's speed and lateral position as continuous predictors, and the mean occupancy normalized transient probability as our dependent variable. Finally, we considered any neuron to be a trajectory-dependent splitter neuron if it had a significant effect of trial type or trial type x stem bin after accounting for speed and lateral position.

**Linear discriminant decoding analysis**. A linear discriminant decoder was trained on data from 50% of trials on a given session using the fitdiscr function in MATLAB. Calcium event activity for each neuron at each time point when the mouse was on the center stem were used as the input variables and the mouse's upcoming turn direction was used as the response variable. Only correct trials were considered for training and testing. The decoder was then used to predict the turn

direction of the other 50% of correct trials, after which the process was repeated 999 times using a different random 50% of trials for training/decoding. The decoding accuracy was then calculated in ~3.3 cm bins along the stem, and the mean accuracy across all bins was taken as the decoding accuracy for that session.

**Functional coding designation**. We first performed neuron registration between all sessions in which the mouse performed more than 20 free trials. We classified neurons as staying active if they were identified by our cell extraction algorithm on both sessions and produced at least five calcium events (while the mouse was running) through the course of the first recording session being considered in the registration. We then categorized cells into five different functional coding types: (1) trajectory-dependent splitter cells, (2) arm place cells, (3) stem place cells, (4) arm non-place cells, and (5) stem non-place cells. Splitter cells were designated based on the criteria listed above. Neurons that produced no calcium activity on the stem of the maze and met our place cell criteria were defined as return arm place cells. Neurons that produced calcium activity on the stem and met our place cell criteria but not our splitter neuron criteria were designated as stem place cells.

**Analysis of neurons that remain active between sessions**. In order to ensure sufficient precision in calculating the probability neurons of a particular functional coding type stayed active, a session-pair was excluded from analysis if there were fewer than four cells in either category in the first session being registered. The probability a neuron class (splitters, stem place cells, or arm place cells) stays active was then calculated as the number of neurons of that class that were active in both sessions divided by the total number of neurons active in the first session. Note that a neuron need not maintain its class between sessions to be considered as active in the second session (how well splitter/place cells maintained their trajectory/place activity is addressed in Fig. 6 and the following "Ontogeny Analysis" section of the Methods). This analysis was performed in two ways: (1) including all cells found for each functional coding type, and (2) matching mean event rate between each functional coding type by excluding the lowest event rate cells for each coding type to match those of splitter neurons. In the event that place cells had a higher mean firing rate than splitter cells, no place cells were excluded. A one-sided sign-test, Holm–Bonferroni corrected for the number of day lags considered (15), was used to determine if splitter neurons were significantly more likely to remain active than Arm PCs or Stem PCs.

**Ontogeny analysis**. We tracked splitter cell ontogeny in three steps. First, we registered all the neurons we recorded across the entire experiment. Second, we identified the first day/session that a neuron passed our statistical criteria to be considered a splitter and defined that session as its onset. Finally, we calculated multiple metrics for the quantity of trajectory-dependent activity produced by each of these neurons (see "Trajectory-Dependent/Splitter Cell Identification" above) in all the sessions preceding and following onset, excluding any sessions that occurred on the same day. The methodology for tracking place cell onset was identical, except mutual information was used as a metric of spatial information provided by each cell.

**Statistics**. Statistical tests used are noted in the corresponding text and figure legends.

**Reporting summary**. Further information on research design is available in the Nature Research Reporting Summary linked to this article.

## Data availability

We have deposited processed imaging and behavioral data at https://doi.org/10.17632/2twf9f834v.1. Raw imaging and behavioral data is available upon reasonable request, contact Nat Kinsky (nat.kinsky@gmail.com).

## Code availability

All custom-written MATLAB code used in this study is freely available at https://github.com/SharpWave/Tenaspis and https://github.com/nkinsky/ImageCamp.

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

## Acknowledgements

First and foremost we would like to thank Dr. Howard Eichenbaum for his help in early stages of this project. We would like to also thank Dr. Jon Rueckemann and Dr. Nick Robinson for input on data analysis, interpretation, and feedback during manuscript preparation. We would like to thank Dr. Steve Gomperts, Dr. Annabelle Singer, Dr. Amar Sahay, Dr. Kamran Diba and the members of their labs for helpful comments during discussions of this work. We would also like to thank Dr. Ian Davison, Dr. Steve Ramirez, Dr. Ehren Newman, and Dr. Mark Kramer, as well as Dr. Andrew Alexander, Dr. Jake Hinman, Dr. John Bladon, Dr. Ryan Place, Dan Sheehan, Winny Ning, Dan Orlin, Jiawen Chen, and Catherine Mikkelsen for feedback and critiques during early stages of data analysis and writing. We would like to thank Annalyse Kohley, Hanish Polavarapu, and Scott Bovino for help with animal training and behavioral tracking quality control. We would like to thank Denise Parisi, Jun Shen, Dr. Shelley Russek, and Sandra Grasso for administrative support. We would like to acknowledge the GENIE Program, specifically Vivek Jayaraman, Ph.D., Douglas S. Kim, Ph.D., Loren L. Looger, Ph.D., Karel Svoboda, Ph.D. from the GENIE Project, Janelia Research Campus, Howard Hughes Medical Institute, for providing the GCaMP6f virus. Finally, we would like to acknowledge Inscopix, Inc. for making single-photon calcium imaging miniscopes widely available, and specifically Lara Cardy and Vardhan Dani for all their technical support throughout and after the experiment. This work was supported by ONR MURI N00014-16-1-2832, NIH R01 MH060013, NIH RO1 MH052090, NIH RO1 MH051570, and NSF NRT UtB: Neurophotonics DGE-1633516.

## Author contributions

Conceptualization: N.R.K.; Methodology: N.R.K.; Software: N.R.K., W.M., D.J.S., S.J.L.; Validation: N.R.K.; Formal Analysis: N.R.K., W.M.; Investigation: N.R.K., W.M.; Resources: N.R.K., M.E.H.; Data Curation: N.R.K., W.M., E.A.R.; Writing—original draft preparation: N.R.K.; Writing—review and editing: N.R.K., W.M., D.J.S., E.A.R., S.J.L., M.E.H.; Visualization: N.R.K., W.M.; Supervision: M.E.H., N.R.K.; Project administration: N.R.K., M.E.H.; Funding Acquisition: M.E.H.

## Competing interests

The authors declare no competing interests.
