## [Peer Review File · Nature Communications]

Reviewers' Comments:

Reviewer #1:

Remarks to the Author:

In this study, Kinsky and colleagues used single photon calcium imaging to examine trajectory-dependent splitter neurons in the mouse hippocampus of mice performing a continuous spatial alternation task. They found that the quality of trajectory dependent activity correlated with performance. Trajectory dependent activity emerged rapidly, persisted across days, and was more stable than place cell fields. On the basis of these results, the authors suggest that hippocampal neurons with useful functional coding properties exhibit greater stability in order to support memory guided behavior. This is an interesting, technically difficult study that attempts to shed light on hard-to-measure cross-session dynamics of an interesting population of cells. Even so, several technical uncertainties could impact interpretation of the results.

1. How do the authors interpret the negative deflections in the example calcium traces of Figure 1E? Also, several of the clearly defined positive deflections evident in this figure appear to have been missed by the event-detection algorithm. Was a threshold of 2 standard deviations above the mean used for event detection, as in Mau et al, 2018?

2. The authors appropriately begin to consider the contribution of differential lateral position on the stem and differential behavioral features in the assessment of putative splitter neurons, but this effort appears to be incomplete.

First, it is not clear why the authors elected to segment the central arm lateral position, rather than treating the stem as a one-dimensional section of the maze. This lateral segmentation increases the likelihood that animal occupancy will vary across the lateral extent of the stem, which could give rise to apparent trajectory splitting due not to neural activity but to behavior. Splitter neurons that meet splitter identification criteria when the central arm is treated as one dimensional would be resilient to this issue.

Second, given the potential impact of behavior in the assessment of putative splitter neurons which may be larger than the authors report, tuning curves along each trajectory should be corrected for occupancy to facilitate splitter cell identification as well as comparison across trajectories and across sessions. This approach appears to have been implemented for place cell detection but not for splitter cells.

Third, the authors point out that a sizeable proportion of putative splitter neurons in each mouse lost their significant modulation by upcoming turn direction after speed and lateral position along the stem were included as covariates. As these neurons are not splitter neurons, they should have been excluded from further analyses.

3. A difference in the field size of splitter cells and place cells, if present, would be expected to contribute to differences between splitter cells and place cells in contrasts of spatial correlations across sessions, such as those of Figure 5. This would be expected to be exacerbated by the small number of calcium events per cell, that may differ by cell type, and potentially by uncertainty in defining the start of a calcium event that could contribute to jitter in place field location for small fields.

4. For the discriminability measure, behavioral differences between left and right turn trials could influence the metric if the tuning curves were not corrected for occupancy. In addition, the use of absolute value complicates the interpretation of discriminability as defined. For the case of a neuron with left-bias on the first half of the stem and an equal amount of right-bias on the second half of the

stem, the discriminability would seem to be high but should be zero if discrimination of trajectory is the objective. Was discriminability instead defined as the absolute value of the sum across bins of the difference between tuning curves?

5. For the reliability measure, reliability would seem to be high for the case described above, in which a neuron has strong left-bias on the first half of the stem and an equal amount of right-bias on the second half of the stem across sessions, yet this cell would provide no net trajectory information. Can the authors confirm that Figure 3 plots reliability for splitter cells alone (since the legend refers to 'all cells')?

6. Was the linear discriminant decoder tested on only correct trials? This would be appropriate given that it was trained on correct trials.

7. It would be helpful to determine whether the size of fields on the stem impacts the measure of 1-rho in Figure 3. Was any smoothing applied to the tuning curves? If so, this would impact the correlation measure.

8. The mouse-level data in Figure 3ABCD suggest that differences between mice, for example on the basis of age, may underlie the significant positive correlations across sessions between performance and the measures of quality of trajectory-dependent activity. Do these correlations persist across sessions within each mouse?

9. In Figure 4, it is not clear how the 'Probability Stays Active' metric is computed. Does the definition of active for each cell class (splitter, stem place cell, arm place cell) require that each cell continue to meet diagnostic criteria for that class across all sessions while firing at least 5 calcium events in each session? For cells of each class defined at the first session, what proportion of cells stay active but change their class in subsequent sessions?

10. It is unclear why splitter neurons differed from arm place cells but not from stem place cells in some analyses, eg Figure S4. The authors raise one possible explanation for this observation, namely that many stem place cells exhibited trajectory-dependent activity insufficient to meet splitter neuron criteria. This could be tested explicitly. There are other explanations that should be considered as well. For example, it remains possible that all neurons active on the stem, by virtue of the stem's relevance to the pending decision, are more likely to stay active across sessions. It is also possible that experimenter movements to provide reward while the animal traversed the return arm could impact return arm calcium activity. The differences between stem cells and arm cells in Figure S5 would seem to support these possibilities.

11. Splitter neurons were defined on the basis of having at least 3 stem bins whose difference in left-right tuning curves survived the shuffle. Did these stem bins need to have the same sign (i.e., all biased to the left, or all biased to the right) to meet this criteria? If not, they would not seem to be trajectory dependent neurons, per se. Did these stem bins need to be spatially clustered along the stem? Did any cell have at least 3 bins biased to left and at least 3 bins biased to right? If so, how were such cases handled?

12. How is 'splitting extent' defined?

13. Methods should be provided explicitly for place cell identification.

14. Since the results of this study are correlative, even if they prove to be consistent with the possibility that hippocampal neurons with useful functional coding properties exhibit greater stability

to support memory guided behavior, they will not provide conclusive evidence.

15. Although not required, comparisons of correct trials to error trials or even to forced trials might have improved understanding of how the splitter neurons contribute to mouse behavior.

Minor points:

In the rasters of figures 2 and 6, what do the horizontal lines depict? Vertical ticks would be preferable to show the onset of calcium event transients. In the associated histograms, the size of the smoothing kernel, if any, should be provided.

In Figure 4B legend, there is an error reference to B.

For what multiple comparisons are the Holm-Bonferroni corrections being employed? The number of spatial bins?

In Figure 6F legend, mean rather than peak discriminability score is stated.

p.52, "See 0 above" is not informative.

Reviewer #2:

Remarks to the Author:

To analyze long-term spatial coding dynamics in the hippocampus of behaving mice, the authors used in vivo miniscopes to assess calcium signals in CA1 neurons as mice performed a continuous alternation task in 30 min sessions recorded over 16-36 days. The study focused on place and "splitter cells:" neurons with place fields whose activity varies with an animal's past or future trajectory. Most previous studies of trajectory coding used microelectrodes to record single unit activity in behaving rats with stability limited to a few days at most. The stability of calcium imaging was used to follow the development and persistence of spatial firing correlates. The new results verified the existence of splitter cells, showed that place and trajectory signals developed rapidly and were stable for days. Moreover, the splitter cells were more stable and predicted both maze location and alternation performance better than "pure" place cells. Together, the results provide new and powerful evidence that the hippocampus "learns" and stably encodes task features made salient by behavior contingencies, neuronal signals that are crucial for memory performance. The results are important, interesting, and compelling, the methods are appropriate, and the manuscript is written clearly and well. I have two suggestions for improving the paper.

The LDA predicted turn direction in each recording session by training 50% of trials and testing on the remaining 50%, with the subsets chosen randomly 1000 times. Presumably the mean of the 1000 training/testing pairs were combined to calculate the mean accuracy for each session, which was then used to calculate Pearson's r . It would be useful to include permutation tests that randomized turning direction to compare ensemble prediction accuracy to chance. This approach dovetails with the one used to assess splitting by single units (lines 975-977, figure 3), and would assess the magnitude of the splitting signal at the level of ensembles.

The authors might want to compare their results with Bahar et al. (Journal of Neuroscience, 2012),

who performed an analogous experiment in rats that compared the development and persistence of place and splitter cells during different types of learning.

Reviewer #3:

Remarks to the Author:

The paper: Persistent trajectory-modulated hippocampal neurons support memory-guided navigation comes out of the excellent lab of Michael Hasselmo. Here they are using a 1-photon miniature microscope to record the spiking activity in the form of calcium transients from large populations of dorsal CA1 neurons. While recorded their mice partake in a spatial alternation task, and the authors see the abrupt emergence of both place fields and trajectory-dependent firing fields. These findings are not new, but the authors track these cells over multiple days and determine the stability of these fields, which has not been done before. They find that Trajectory-dependent firing fields come online later than place fields over a timescale of days, and trajectory-dependent firing typically appears in pre-established place fields. In addition, they show that performance in the alternation task is correlated with reliability and discriminability of trajectory-dependent firing fields, and a trained decoder can somewhat predict performance based on trajectory-dependent cell activity.

The paper is very well written, and the data support their claims. It is of interest to a broad audience and I'm sure will be well received by the field. I only have one major concern that needs addressing, and a couple of minor ones.

Major

One concern I have is their ability to measure from the same neurons over such long time windows (15 days) without potential confounds through changes in GCaMP expression levels which alters calcium kinetics and adversely affects cell health. It is possible that things are somewhat stable over a couple of weeks, but I would like to see some quantification of this so that we can be confident in the results. 2 features that can be easily measured are baseline fluorescence and calcium transient kinetics. This can be done for each neuron extracted from their fields of view and plotted over days. The authors use a viral construct which has synapsin as the promoter for GCaMP expression, and they inject a relatively high volume (250 nl). This virus has the tendency to ramp up expression quickly and also lead to over-expression relatively quickly versus other promoters of GCaMP such as CaMKII. This is another reason for the authors to show that expression level changes over time aren't affecting their data.

Along these lines, it states in their methods that they inject virus 3-4 weeks prior to implantation, and then another 2 weeks before attaching a base plate. That's 6 weeks of GCaMP expression before they image the CA1, and then another 2 weeks of imaging puts them out to 8 weeks. In our hands syn-GCaMP6 reaches a peak after about 3 weeks, and things look over expressed after 6 weeks. I think it is important to confirm in this study that expression levels, cell health, transient kinetics etc. are within normal levels and are not changing over the course of their experiments.

Minor

1. I have slight small issue with the title. The authors don't show that trajectory-dependent neurons support memory guided behavior, they show the activity of these neurons correlates with task performance. If they want to say "support" they would need to manipulate these neurons specifically, and show task performance is disrupted. Instead of doing that, I recommend the authors change the wording of the title to better reflect their findings.

2. The authors state in the text on line 136: "that in many cases these neurons maintained the same functional phenotype across multiple days". They again state this notion similarly in line 151: "in many cases maintains the same activity profile across both short and long timescales." But the authors only show 2 example cells in Fig 2. They should show a summary plot of all of their trajectory neurons and whether they are still trajectory neurons over days.

3. Figure 6 shows when splitter cells first come online over a timescale of days. It would also be interesting if the authors analyzed on which trial of the maze the firing first appeared on a particular day. For instance, on the day they first came online were the firing fields present on the first trial? Or is there variability on when trial they appear? This analysis would be a nice addition to the paper and would reveal whether these changes over days occur in between sessions or within sessions. This finding could also be placed in the context of a recent paper in *Neuron* that measured on which laps place fields appeared in both familiar and novel environments and found that many place fields appear instantly on lap 1, and many others appear on later laps distributed between laps 2-15 (Sheffield et. al. 2017).

4. In the discussion from line 395 onwards the authors talk about turnover of hippocampal neuron activity over days that has been shown in a number of labs using 1 photon imaging. To bolster their argument further, they should include a recent paper in *Nature* from the Bartos lab (Hainmueller & Bartos, 2018) which shows a similar turnover in CA1 but they use 2P imaging, which doesn't have the inherent problems of z-drift that you get with the 1 photon mini-microscope, as the imaging plane can be matched up on each imaging day.

General Comments: We thank the reviewers for their thoughtful comments. The resulting manuscript is clearer and more complete as a result. We believe the revised manuscript provides a comprehensive description of the development and persistence of trajectory-dependent activity across days. Please see below for specific responses to each comment.

Reviewer #1 (Remarks to the Author):

In this study, Kinsky and colleagues used single photon calcium imaging to examine trajectory-dependent splitter neurons in the mouse hippocampus of mice performing a continuous spatial alternation task. They found that the quality of trajectory dependent activity correlated with performance. Trajectory dependent activity emerged rapidly, persisted across days, and was more stable than place cell fields. On the basis of these results, the authors suggest that hippocampal neurons with useful functional coding properties exhibit greater stability in order to support memory guided behavior. This is an interesting, technically difficult study that attempts to shed light on hard-to-measure cross-session dynamics of an interesting population of cells. Even so, several technical uncertainties could impact interpretation of the results.

1. How do the authors interpret the negative deflections in the example calcium traces of Figure 1E? Also, several of the clearly defined positive deflections evident in this figure appear to have been missed by the event-detection algorithm. Was a threshold of 2 standard deviations above the mean used for event detection, as in Mau et al, 2018?

This is a great question. The cell detection algorithm (Tenaspis) used in this manuscript has undergone several updates since Mau et al. (2018) and now utilizes an adaptive threshold rather than the 2 standard deviation threshold used in Mau et al. (2018). This adaptive threshold better accounts for differences in fluorescence level and signal-to-noise ratio between neurons. This version also employs a spatial band-pass filter used by others (Sun et al., 2015) to help disambiguate adjacent neurons. When this band-pass filter is applied it suppresses pixel intensity in a small ring surrounding the active neuron: this helps prevent cross-talk in a neighboring neuron region-of-interest (ROI) by limiting bleed-through of fluorescence from one neuron to the other. The end result is that when one neuron is inactive but an adjacent neuron has a calcium event this appears as a small downward deflection in that neuron's ROI.

That being said, in some cases (e.g. if a neighboring ROI is highly overlapping, as is the case with the top-most purple neuron in Figure 1E) fluorescence from one ROI can still invade another. This can appear to be a missed-event in the raw traces. However, these missed events are actually fluorescent activity that has been correctly attributed to a neighboring (typically highly fluorescent) neuron ROI. We have revised this figure in the updated manuscript to illustrate how our algorithm disambiguates activity in overlapping neurons (also shown below). For the two overlapping neurons shown, the smaller transients that appear to be missed events result from fluorescence originating in (and correctly attributed to) the neighboring neuron (C below). The enlarged frames from the peak of each transient (D below) clearly show the fluorescence centered in the appropriate neuron. For example, the event at time 2 is attributed to the top trace, and the event at time 4 is attributed to the bottom trace. We have added the figure subpanels shown below to Supplemental Figure 1.

2. The authors appropriately begin to consider the contribution of differential lateral position on the stem and differential behavioral features in the assessment of putative splitter neurons, but this effort appears to be incomplete.

First, it is not clear why the authors elected to segment the central arm lateral position, rather than treating the stem as a one-dimensional section of the maze. This lateral segmentation increases the likelihood that animal occupancy will vary across the lateral extent of the stem, which could give rise to apparent trajectory splitting due not to neural activity but to behavior. Splitter neurons that meet splitter identification criteria when the central arm is treated as one dimensional would be resilient to this issue.

Our analysis in both the previous and current submission segments the stem into five bins along its length but treats the center stem as one dimensional with lateral position treated as a continuous predictor in line with Wood et al. (2000). The methods section has been revised as follows to clarify (underlining to highlight the changes):

“In order to check the robustness of our results and control for any trajectory-dependent information resulting from stereotyped deviations in speed or lateral position along the stem, we also performed an additional analysis in line with previous studies (Ito et al., 2015; Wood et al., 2000). To do so, we first divided the stem lengthwise into five bins and calculated the average transient probability in each bin for all trials. Note that we did not segment the stem laterally but instead used lateral position as a continuous predictor (as described in the following sentences) in line with Wood et al. (2000). We then performed an ANOVA analysis using the anovan function in MATLAB for each trajectory-dependent splitter neuron we detected. We used trial type (left/right), stem bin, stem bin x trial type as categorical predictors, the animal’s speed and lateral position as continuous predictors, and the mean occupancy normalized calcium event probability as our dependent variable. Finally, we considered any neuron to be a trajectory-dependent splitter neuron if it had a significant effect of trial type or trial type x stem bin after accounting for speed and lateral position.”

Second, given the potential impact of behavior in the assessment of putative splitter neurons which may be larger than the authors report, tuning curves along each trajectory should be corrected for occupancy to facilitate splitter cell identification as well as comparison across trajectories and across sessions. This approach appears to have been implemented for place cell detection but not for splitter cells.

In both the previous and current submission, our analysis of splitter cells does correct for occupancy along the stem (as with place cells). We have clarified how we calculated tuning curves in the methods (underlining to highlight changes):

“...After parsing the animal’s behavior into these sections, the center stem was broken up into ~1cm bins and the event rate for each neuron was calculated for each trial. Occupancy normalized tuning curves for each trial type (left or right turn) were then constructed, which consisted of each neuron’s mean event rate for all correct trials at each spatial bin divided by the time the mouse spent in each bin. The difference between these curves was then calculated....”

Third, the authors point out that a sizeable proportion of putative splitter neurons in each mouse lost their significant modulation by upcoming turn direction after speed and lateral position along the stem were included as covariates. As these neurons are not splitter neurons, they should have been excluded from further analyses.

We agree. In response to this comment, we have re-run all analyses excluding any neurons that lose their trajectory modulation after accounting for speed/lateral position. All figures utilizing the splitter designation were likewise revised to reflect the exclusion of these non-trajectory modulated neurons. Note that we also conservatively excluded a number of neurons to address reviewer #3’s comment about the potential effects of long-term GCaMP expression and that all figures in the revised manuscript now reflect both exclusion criteria. After this re-analysis, Figures 1-3 remain similar to the initial submittal, as do Figures 5-7, though the breadth of statistical significance has been slightly reduced in Figure 3. The likelihood that splitters remain active compared to stem place cells shown in Figure 4 has diminished. However, based on reviewer comments, we now provide a more complete picture of different cell type activity across days in Figure 4: splitter neurons are more likely than arm place cells, but not stem place cells, to remain active across sessions. In addition, splitter neurons display a stronger difference from non-place cells in their tendency to remain active, which supports the idea of reviewer #1 and the work of Bahar et al. (2012) that all neurons providing relevant (place and trajectory) information for the upcoming decision require stability.

3. A difference in the field size of splitter cells and place cells, if present, would be expected to contribute to differences between splitter cells and place cells in contrasts of spatial correlations across sessions, such as those of Figure 5. This would be expected to be exacerbated by the small number of calcium events per cell, that may differ by cell type, and potentially by

uncertainty in defining the start of a calcium event that could contribute to jitter in place field location for small fields.

Splitter cells do have larger fields than stem place cells, and stem place cells have larger fields than arm place cells. We have acknowledged this in figure S5A, shown below. However, place field size does not entirely account for differences in place field correlations since stem place cells have significantly larger place fields than arm place cells yet also have slightly lower (but not statistically significant) place field correlations between sessions (see S5B below):

4. For the discriminability measure, behavioral differences between left and right turn trials could influence the metric if the tuning curves were not corrected for occupancy.

We did correct for occupancy (see our response to comment #2 above) so behavioral differences between different turn directions should not influence this metric.

In addition, the use of absolute value complicates the interpretation of discriminability as defined. For the case of a neuron with left-bias on the first half of the stem and an equal amount of right-bias on the second half of the stem, the discriminability would seem to be high but should be zero if discrimination of trajectory is the objective. Was discriminability instead defined as the absolute value of the sum across bins of the difference between tuning curves?

This is a good question. The methods section is correct: discriminability was defined as the sum of the absolute value of the difference between left and right tuning curves across all bins. If a downstream brain region does indeed use hippocampal signal to guide behavior, it could reliably decode the animal's prior trajectory from the activity of the example neuron described above at any point along the stem (high firing at the beginning of the stem = go right, high firing at the end = go left). Thus, high discriminability, even in the case outlined above where there is no net bias, would still allow for accurate discrimination of trajectory at each point along the stem. We acknowledge that no one term or metric can completely capture the many ways in which neurons exhibit trajectory-dependent activity on the stem of the maze. We use discriminability and reliability to describe, to the best of our ability, distinct attributes of trajectory-dependent activity: the *magnitude* and *consistency* (see our response to the next comment) of differences between left and right turn activity rates.

5. For the reliability measure, reliability would seem to be high for the case described above, in which a neuron has strong left-bias on the first half of the stem and an equal amount of right-bias on the second half of the stem across sessions, yet this cell would provide no net trajectory information.

Reliability was defined to assess the consistency of trajectory-dependent activity. A neuron that never fired on left turn trials and but fired robustly on 50% of right turn trials would have high discriminability but relatively low reliability. A neuron that consistently produced three transients on each left turn trial and one transient on each right turn trial might have similar discriminability to the first neuron but would have much higher reliability.

Can the authors confirm that Figure 3 plots reliability for splitter cells alone (since the legend refers to 'all cells')?

Figure 3 includes all cells that are active on the central stem. We have clarified the Figure 3 legend to state that the plot includes all cells that are active on the stem, not just splitters. This analysis cannot be done with splitters alone since, by definition, splitters have very high reliability and discriminability values. Including other neurons active on the stem allows us to assess whether a shift of the entire neuron population toward better/more reliable discrimination between left and right trajectories correlates with performance.

6. Was the linear discriminant decoder tested on only correct trials? This would be appropriate given that it was trained on correct trials.

Confirmed. This has been clarified in the methods section: "...Only correct trials were considered for training and testing. The decoder was then used to predict the turn direction of the other 50% of correct trials..."

7. It would be helpful to determine whether the size of fields on the stem impacts the measure of 1-rho in Figure 3. Was any smoothing applied to the tuning curves? If so, this would impact the correlation measure.

No smoothing was applied to tuning curves: correlations were calculated based on unsmoothed tuning curves in ~1cm bins. We have clarified this in the methods: "...Last, we calculated the correlation between left and right unsmoothed tuning curves (~1cm bins)..."

8. The mouse-level data in Figure 3ABCD suggest that differences between mice, for example on the basis of age, may underlie the significant positive correlations across sessions between performance and the measures of quality of trajectory-dependent activity. Do these correlations persist across sessions within each mouse?

Positive correlations persist across 3 of the 4 mice for mean reliability and mean discriminability, and are significant for one mouse for reliability and close to significant for one mouse for discriminability. We thus pooled data across animals to obtain sufficient statistical power to calculate these correlations on a session by session basis. After revision (in particular excluding neurons with long calcium kinetics per reviewer #3 comments) the correlation between LDA accuracy and performance as well as between 1- rho and performance have diminished and now approach significance at the level of individual mice only. Not surprisingly, we do not observe consistent correlations in these metrics at the individual mouse level, and believe this is reflected in the group data plots of Figure 3C-D.

9. In Figure 4, it is not clear how the 'Probability Stays Active' metric is computed. Does the definition of active for each cell class (splitter, stem place cell, arm place cell) require that each cell continue to meet diagnostic criteria for that class across all sessions while firing at least 5 calcium events in each session? For cells of each class defined at the first session, what proportion of cells stay active but change their class in subsequent sessions?

This is a great question. The probability a neuron stays active does not require it to maintain its class. We have added the following text to the methods section to clarify: "The probability a neuron class (splitters, stem place cells, or arm place cells) stays active was then calculated as the number of neurons of that class that were active in both sessions divided by the total number of neurons active in the first session. Note that a neuron need not maintain its class between sessions to be considered as active in the second session (how well splitter/place cells maintained their trajectory/place activity is addressed in Figure 6 and the following "Phenotype Ontogeny Analysis" section of the Methods. We observed no clear trends in the likelihood that each neuron maintained its class (y-axis below) between sessions as none of the distributions shown below (for sessions 1 day apart) are significantly different:

10. It is unclear why splitter neurons differed from arm place cells but not from stem place cells in some analyses, eg Figure S4. The authors raise one possible explanation for this observation, namely that many stem place cells exhibited trajectory-dependent activity insufficient to meet splitter neuron criteria. This could be tested explicitly.

This is a good idea. We have added Figure S4G-H, which compares the probability of staying active for high reliability stem PCs (neurons active on the stem with close to, but not quite, significant trajectory-dependent activity) versus low reliability stem PCs. We show that highly reliable stem PCs are more likely to stay active than low reliability stem PCs, though similar to our other analyses, this difference goes away when we match event-rates between neuron pools.

There are other explanations that should be considered as well. For example, it remains possible that all neurons active on the stem, by virtue of the stem's relevance to the pending decision, are more likely to stay active across sessions.

Figure 4 and S4 have been revised after excluding speed/lateral position modulated neurons and neurons with abnormally long calcium kinetics. As a result, splitter neurons are no longer more likely to remain active than stem place cells (Figure 4D), supporting the possibility raised by reviewer #1 above. However, they are more likely to remain active than non-place cells which are active on the stem (Figure 4E-F). We have added

the following sentence to acknowledge this point: "...However, we found that splitters and stem place cells were equally likely to remain active at all time-lags (Error! Reference source not found.E-F). This could occur because stem place cells also exhibit trajectory-dependent activity that does not quite meet our stringent splitter neuron criteria. In support of this idea, stem cells carrying highly reliable trajectory information were more likely to remain active than those carrying relatively unreliable trajectory information (Figure S4G-H). Alternatively this finding could support the idea that both the animal's current and past position are relevant for task performance, which in turn could influence the stability of place cells and splitters, respectively (Bahar & Shapiro, 2012). ..."

It is also possible that experimenter movements to provide reward while the animal traversed the return arm could impact return arm calcium activity. The differences between stem cells and arm cells in Figure S5 would seem to support these possibilities.

This is definitely a possibility. However, in the revised manuscript (after excluding neurons with abnormally long calcium kinetics), across-day spatial correlations are no different between arm place cells and stem place cells (Figure S5B), even though stem place cells have larger place fields than arm place cells. This suggests that place field size or experimenter movements do not entirely account for the difference in correlations between cell groups. This is acknowledged in the revised manuscript: "The higher correlations for splitters were not explained by their place-field size nor by experimenter movements to provide reward on the return arm since stem place cells also have larger place fields, but not spatial correlations, than arm place cells (Figure S5A-B)."

11. Splitter neurons were defined on the basis of having at least 3 stem bins whose difference in left-right tuning curves survived the shuffle. Did these stem bins need to have the same sign (i.e., all biased to the left, or all biased to the right) to meet this criteria? If not, they would not seem to be trajectory dependent neurons, per se. Did these stem bins need to be spatially clustered along the stem? Did any cell have at least 3 bins biased to left and at least 3 bins biased to right? If so, how were such cases handled?

Stem bins for a given neuron did not have to have the same sign in order for that neuron to be considered a splitter. We chose to include these neurons since even neurons with no net bias can still contain information about the mouse's trajectory, and since we don't know at which point on the stem the mouse decides he is going a particular direction nor do we know at which point any downstream structures might utilize information from hippocampus neurons to begin guiding behavior.

Consider a theoretical neuron which fired the same amount on each trial, but at the beginning of the stem on left trials and at the end of the stem on right trials. Based on that single neuron, a theoretical downstream structure could reliably decode if the mouse was on a right-turn trajectory if that neuron was firing at the end of the stem, despite it having no net bias. Nevertheless, cases where a splitter did not exhibit a net bias toward one direction (# left biased bins - # right biased bins = 0) were infrequent: only 13 of 2662 splitters recorded exhibited no net bias, defined as having less than 3 net bins in one direction. Bins did not have to be spatially clustered for that neuron to be considered a splitter: our stringent criteria sometimes resulted in 1-2 consecutive

significant bins separated by a gap even for neurons with clear trajectory-modulated activity (see two example neurons below).

Also, see additional example splitter neurons across days shown in Figure S2E-N. Most splitter neurons increased event rate for one trial type: those that shifted their activity along the stem were rare.

12. How is 'splitting extent' defined?

Splitting extent is defined as the proportion of the stem where there was a significant difference between left and right tuning curves. We have explicitly mentioned this in the text pertaining to Figure 6: "To address this question, we identified the day when each neuron we recorded first exhibited significant trajectory-dependent activity and tracked its splitting extent - the proportion of the stem which exhibited significant differences between left and right tuning curves - in subsequent sessions." We have also added this definition into the methods section: "Finally, we defined splitting extent for each neuron as the proportion of stem bins that exhibited significant differences between left and right tuning curves."

13. Methods should be provided explicitly for place cell identification.

We have quoted the relevant section from Kinsky et al. (2018) since we utilize the same methods and MATLAB code to identify place cells as in that study. Additionally, we have added a paragraph describing how we calculate place field size in response to comment #3 in the methods section under "Place Cell Identification" (lines 1056-1083)

14. Since the results of this study are correlative, even if they prove to be consistent with the possibility that hippocampal neurons with useful functional coding properties exhibit greater stability to support memory guided behavior, they will not provide conclusive evidence.

This is true. We have chosen our language carefully throughout to make this clear (underlining for emphasis): “These findings combined support the idea that the task relevance of information carried by a neuron influences its likelihood to maintain activity at later time points, which could be exploited for successful memory-guided behavior across days.” “This indicates that trajectory-dependent splitter neurons might guide memory task performance by providing a more consistent representation of space than place cells.” Importantly, we have also adjusted the title to: “Trajectory-modulated hippocampal neurons persist throughout memory-guided navigation” Which we believe accurately describes the nature of our results without implying that we causally tested this relationship.

15. Although not required, comparisons of correct trials to error trials or even to forced trials might have improved understanding of how the splitter neurons contribute to mouse behavior.

We were also interested in this question but ultimately decided not to include it in this manuscript due to the large behavioral differences we encountered between correct trials and forced/error trials. For example, during forced trials mice tended to hug the sides of the center stem rather than running down the middle (left trials in blue, right trials in green).

Looping

During correct trials, in contrast, mice tended to stick to the middle and run straight down the stem (notice the highly overlapping blue and green trajectories below and note that we only included the middle ~60-70% of the stem in our analysis to eliminate any points where trajectories began to diverge). On error trials (shown in red below), mice tended to backtrack, spin around, and generally spend much more time on the stem than during correct trials (there are many circuitous paths on the stem below in red). The single backtracking path shown in green below is the exception that proves the rule.

Correct

Error

Finally, in many sessions the mice only exhibited perseveration errors, which does not allow for analysis of trajectory-dependent activity since only one trajectory was taken.

Correct

Error

Ultimately, we decided that the large differences between behavior on correct trials and forced/error trials did not permit for a fair or interpretable comparison with correct trial data.

Minor points:

In the rasters of figures 2 and 6, what do the horizontal lines depict? Vertical ticks would be preferable to show the onset of calcium event transients. In the associated histograms, the size of the smoothing kernel, if any, should be provided.

The horizontal lines depict the rising phase of that neuron's calcium trace which corresponds with spiking activity in neurons (see Chen et al., Nature, 2013). Horizontal lines are used since sampling rate (20 frames/second) and GCaMP kinetics are relatively slow compared to electrophysiology. Thus, we are representing the full extent of the rising phase, which is when spiking activity occurs in neurons.

The tuning curves in both figures were smoothed using the fit function in MATLAB with a smoothing spline ($p=0.9$). This has been noted in the figure legend and methods.

In Figure 4B legend, there is an error reference to B.

Good catch. This has been fixed and now refers to A.

For what multiple comparisons are the Holm-Bonferroni corrections being employed? The number of spatial bins?

The Holm-Bonferroni correction is conservatively applied to the number of day lags we considered (15). We have clarified this in the figure legend (“...after Holm-Bonferroni correction (15 day lags considered)...”) and methods sections (“...A one-sided sign-test, Bonferroni-Holm corrected for the number of day lags considered (15), was used to

determine if splitter neurons were significantly more likely to remain active than Arm PCs or Stem PCs...”).

In Figure 6F legend, mean rather than peak discriminability score is stated.

Good catch – this has been corrected.

p.52, “See 0 above” is not informative.

Another good catch. This was supposed to state “see Trajectory-Dependent/Splitter Cell Identification above” and has been fixed.

Reviewer #2 (Remarks to the Author):

To analyze long-term spatial coding dynamics in the hippocampus of behaving mice, the authors used in vivo miniscopes to assess calcium signals in CA1 neurons as mice performed a continuous alternation task in 30 min sessions recorded over 16-36 days. The study focused on place and “splitter cells:” neurons with place fields whose activity varies with an animal’s past or future trajectory. Most previous studies of trajectory coding used microelectrodes to record single unit activity in behaving rats with stability limited to a few days at most. The stability of calcium imaging was used to follow the development and persistence of spatial firing correlates. The new results verified the existence of splitter cells, showed that place and trajectory signals developed rapidly and were stable for days. Moreover, the splitter cells were more stable and predicted both maze location and alternation performance better than “pure” place cells. Together, the results provide new and powerful evidence that the hippocampus “learns” and stably encodes task features made salient by behavior contingencies, neuronal signals that are crucial for memory performance. The results are important, interesting, and compelling, the methods are appropriate, and the manuscript is written clearly and well. I have two suggestions for improving the paper.

The LDA predicted turn direction in each recording session by training 50% of trials and testing on the remaining 50%, with the subsets chosen randomly 1000 times. Presumably the mean of the 1000 training/testing pairs were combined to calculate the mean accuracy for each session, which was then used to calculate Pearson’s r . It would be useful to include permutation tests that randomized turning direction to compare ensemble prediction accuracy to chance. This approach dovetails with the one used to assess splitting by single units (lines 975-977, figure 3), and would assess the magnitude of the splitting signal at the level of ensembles.

Good suggestion. We have performed this analysis and now show the mean of the shuffled LDA decoding values on the plots in Figure 3. In general, the decoder performs well above chance (~50%), as long as the mouse’s behavioral performance is also high.

The authors might want to compare their results with Bahar et al. (Journal of Neuroscience, 2012), who performed an analogous experiment in rats that compared the development and persistence of place and splitter cells during different types of learning.

This is a great suggestion: Bahar et al. (2012) is highly relevant to this study. In particular, the finding that stable place cell *and* trajectory-dependent activity is required for proper task performance could explain why we observe correlations between performance and some trajectory-dependent metrics but not others in Figure 3. We have cited this study in numerous places throughout the revised manuscript.

Anecdotally, we do observe a correlation between the proportion of cells exhibiting trajectory-dependent activity (splitter proportion) and performance, in agreement with Bahar (2012). However, we also observed a strong correlation between splitter proportion and number of trials performed (we did not have our mice perform the same number of trials each day but rather placed a time cap on each session for concerns of photobleaching), as well as between performance and number of trials performed. Thus, it is unclear if the increase in splitter proportion with better performance is real or if it is driven by increased statistical power resulting from more trials.

Reviewer #3 (Remarks to the Author):

The paper: Persistent trajectory-modulated hippocampal neurons support memory-guided navigation comes out of the excellent lab of Michael Hasselmo. Here they are using a 1-photon miniature microscope to record the spiking activity in the form of calcium transients from large populations of dorsal CA1 neurons. While recorded their mice partake in a spatial alternation task, and the authors see the abrupt emergence of both place fields and trajectory-dependent firing fields. These findings are not new, but the authors track these cells over multiple days and determine the stability of these fields, which has not been done before. They find that Trajectory-dependent firing fields come online later than place fields over a timescale of days, and trajectory-dependent firing typically appears in pre-established place fields. In addition, they show that performance in the alternation task is correlated with reliability and discriminability of trajectory-dependent firing fields, and a trained decoder can somewhat predict performance based on trajectory-dependent cell activity.

The paper is very well written, and the data support their claims. It is of interest to a broad audience and I'm sure will be well received by the field. I only have one major concern that needs addressing, and a couple of minor ones.

Major

One concern I have is their ability to measure from the same neurons over such long time windows (15 days) without potential confounds through changes in GCaMP expression levels which alters calcium kinetics and adversely affects cell health. It is possible that things are somewhat stable over a couple of weeks, but I would like to see some quantification of this so that we can be confident in the results. 2 features that can be easily measured are baseline fluorescence and calcium transient kinetics. This can be done for each neuron extracted from their fields of view and plotted over days. The authors use a viral construct which has synapsin as the promoter for GCaMP expression, and they inject a relatively high volume (250 nl). This virus has the tendency to ramp up expression quickly and also lead to over-expression relatively quickly versus other promoters of GCaMP such as CaMKII. This is another reason for the

authors to show that expression level changes over time aren't affecting their data.

Along these lines, it states in their methods that they inject virus 3-4 weeks prior to implantation, and then another 2 weeks before attaching a base plate. That's 6 weeks of GCaMP expression before they image the CA1, and then another 2 weeks of imaging puts them out to 8 weeks. In our hands syn-GCaMP6 reaches a peak after about 3 weeks, and things look over expressed after 6 weeks. I think it is important to confirm in this study that expression levels, cell health, transient kinetics etc. are within normal levels and are not changing over the course of their experiments.

In order to limit overexpression to the best of our ability, we injected a relatively small amount and titer compared to other studies using virally expressed GCaMP (Cai et al., 2016 - 500nL; Grewe et al., 2017 - 500nL; Krabbe et al., 2019 - 400nL; Rubin et al., 2015 - 400nL; Zhou et al., 2019 - 1uL; Ziv et al., 2013 - 250nL). Of the studies listed above that used the synapsin promoter, the titer we use is ~50% of that listed in Zhou et al. (Cai et al. also used the synapsin promoter but does not list the titer they inject to the best of our knowledge). To assess stability of our viral expression, we have quantified both the cumulative distribution of half-life of neurons (top row) and the cumulative distribution of their baseline fluorescence levels (bottom row) over each week of recording in Figure S1E-F, as shown here (different colors indicate different weeks). Note that the mice in this study also underwent several weeks of habituation to different environments and to having the camera attached, and that two of the mice participated in another study prior to this one. As a result, the mean time the virus was in the system on the last imaging day was 19.4 weeks. This has been noted in the methods section: "All mice were imaged in open arenas for several weeks to habituate them to attaching and wearing the camera (see following section). Additionally, two of the mice participated in another experiment prior to imaging. As a result, the mean time the virus was in the system on the last imaging day was 19.4 weeks."

We see no evidence of systematic increases in either metric, suggesting stable expression of our calcium indicator. Nonetheless, we have opted to exclude any neurons with abnormally long transients (> 2 seconds) based on an upper limit observed in Lian

et al. (2009). This is acknowledged in the methods section and in the text (lines 115-118): “We observed no systematic increases in calcium trace kinetics or fluorescence across sessions, indicating stable levels of GCaMP expression (Error! Reference source not found.E-F). Nevertheless, we excluded any potentially unhealthy neurons that had half-decay times > 2 seconds ($13\% \pm 6.7\%$ of neurons across all sessions).” This is not to say that no mice in our study exhibited overexpression – a number of mice were prepared and screened for this study but were never trained on the task due to continually increasing fluorescence and aberrant activity. Only those mice that we observed with stable appearing fluorescence levels were trained on the task and included in the study. As a side note, we found little to no documentation of calcium kinetics and/or fluorescence levels across time (Ziv et al. 2013 is one exception) in the existing articles in the literature, especially for single-photon calcium imaging, so we believe setting a precedent of documenting these values is a good idea for the field in general.

In summary, in response to this comment and comment #3 by reviewer #1, we have revised the manuscript after a) excluding a subset of neurons with abnormally long calcium kinetics from consideration and b) re-classifying any putative splitter neurons that lose their modulation by turn-direction after accounting for speed and lateral position. After this re-analysis, Figures 1-3 remain similar to the initial submittal, as do Figures 5-7, though the breadth of statistical significance has been slightly reduced in Figure 3. The likelihood that splitters remain active compared to stem place cells shown in Figure 4 has diminished. However, based on reviewer comments, we now provide a more complete picture of different cell type activity across days in Figure 4: splitter neurons are more likely than arm place cells, but not stem place cells, to remain active across sessions. In addition, splitter neurons display a stronger difference from non-place cells in their tendency to remain active, which supports the idea of reviewer #1 and the work of Bahar et al. (2012) that all neurons providing relevant (place and trajectory) information for the upcoming decision require stability.

Minor

1. I have slight small issue with the title. The authors don't show that trajectory-dependent neurons support memory guided behavior, they show the activity of these neurons correlates with task performance. If they want to say “support” they would need to manipulate these neurons specifically, and show task performance is disrupted. Instead of doing that, I recommend the authors change the wording of the title to better reflect their findings.

We agree. To account for this fact, we have adjusted the title to: “Trajectory-modulated hippocampal neurons persist throughout memory-guided navigation.”

2. The authors state in the text on line 136: “that in many cases these neurons maintained the same functional phenotype across multiple days”. They again state this notion similarly in line 151: “in many cases maintains the same activity profile across both short and long timescales.” But the authors only show 2 example cells in Fig 2. They should show a summary plot of all of their trajectory neurons and whether they are still trajectory neurons over days.

This is a good suggestion which we have incorporated into the manuscript (due to constraints on the size of figures this was placed in supplemental Figure S2C-D). We

have also provided a complementary summary plot of splitting extent which provides a nice visual confirmation of our group data shown in Figure 6D. Thank you. Finally, we have provided twenty more examples of splitter neurons the majority of which stayed splitters, from 1-10 days later in Figure S2E-N. These examples of splitter neurons across days in Figure S2 provide numerous visual confirmations of the high consistency of spatial and trajectory-dependent activity shown in Figures 5-6.

3. Figure 6 shows when splitter cells first come online over a timescale of days. It would also be interesting if the authors analyzed on which trial of the maze the firing first appeared on a particular day. For instance, on the day they first came online were the firing fields present on the first trial? Or is there variability on when trial they appear? This analysis would be a nice addition to the paper and would reveal whether these changes over days occur in between sessions or within sessions. This finding could also be placed in the context of a recent paper in Neuron that measured on which laps place fields appeared in both familiar and novel environments and found that many place fields appear instantly on lap 1, and many others appear on later laps distributed between laps 2-15 (Sheffield et. al. 2017).

Good suggestion. We have performed this analysis and incorporated the results into Figure 7E-F which shows the time course of splitter or place cell appearance across both time and trials for one mouse, shown below, and all mice combined (insets). Like Sheffield (2017) we also find that a large proportion of neurons become active early in the session/on the first two trials. However, in contrast to our across day results, splitters generally become active before place cells on a given session.

4. In the discussion from line 395 onwards the authors talk about turnover of hippocampal neuron activity over days that has been shown in a number of labs using 1 photon imaging. To bolster their argument further, they should include a recent paper in Nature from the Bartos lab (Hainmueller & Bartos, 2018) which shows a similar turnover in CA1 but they use 2P imaging, which doesn't have the inherent problems of z-drift that you get with the 1 photon mini-microscope, as the imaging plane can be matched up on each imaging day.

This is a great suggestion – it definitely strengthens our point. We have incorporated the following sentence into the revised manuscript: “Third, a notable recent study found similar rates of turnover in CA1 to what we observe using two-photon imaging, which mitigates any concerns of z-plane drift (Hainmueller & Bartos, 2018).”

Reviewers' Comments:

Reviewer #1:

Remarks to the Author:

The authors have responded adequately to the issues raised.
I have no further concerns.

Reviewer #2:

Remarks to the Author:

I liked the first version of this paper, and like the revision more. The authors' responses to reviews are thoughtful and appropriate, and the results of the new and improved analyses make what was already a compelling story yet stronger.

I have minor suggestions (that the authors and editors should feel free to ignore) for the discussion about splitter cells in relation to hippocampus-dependent memory function. In the paragraph starting on page 23, line 457, the discussion describes Ito's paper, in which silencing the n. reuniens reduced CA1 journey coding without affecting task performance. Given, as noted in the revised manuscript, that the task is hippocampus independent, intact performance in that case provides no evidence regarding the importance of splitter cell activity to memory function. Memory tasks that do require hippocampal function are impaired by disrupting nucleus reuniens (e.g. Viena et al., Hippocampus 2018), and the magnitude of CA1 splitting is reduced by inactivating medial prefrontal cortex (Guise et al., Neuron 2017), suggesting that splitter activity may well be important for memory. Along the same lines, the paragraph on page 24-25, lines 478-496, should emphasize that hippocampal dysfunction impairs performance of the discrete trial spatial memory tasks in which splitter cell activity predicted performance (e.g. Ferbinteanu, et al., 2003).

Reviewer #3:

Remarks to the Author:

The authors did a thorough job in addressing my original concerns, which I think has enhanced the paper and increased its transparency. I have no additional comments and feel it is ready for publication and will be a very nice addition to the field.

General Comments: We have incorporated the suggestions of Reviewer 2 into the final version of the manuscript – see below for our detailed response.

REVIEWERS' COMMENTS:

Reviewer #1 (Remarks to the Author):

The authors have responded adequately to the issues raised.

I have no further concerns.

Reviewer #2 (Remarks to the Author):

I liked the first version of this paper, and like the revision more. The authors' responses to reviews are thoughtful and appropriate, and the results of the new and improved analyses make what was already a compelling story yet stronger.

I have minor suggestions (that the authors and editors should feel free to ignore) for the discussion about splitter cells in relation to hippocampus-dependent memory function. In the paragraph starting on page 23, line 457, the discussion describes Ito's paper, in which silencing the n. reuniens reduced CA1 journey coding without affecting task performance. Given, as noted in the revised manuscript, that the task is hippocampus independent, intact performance in that case provides no evidence regarding the importance of splitter cell activity to memory function. Memory tasks that do require hippocampal function are impaired by disrupting nucleus reuniens (e.g. Viena et al., Hippocampus 2018), and the magnitude of CA1 splitting is reduced by inactivating medial prefrontal cortex (Guise et al., Neuron 2017), suggesting that splitter activity may well be important for memory.

Good suggestion. We have added the following lines to the discussion:

“Finally, other recent studies utilizing hippocampal-dependent memory tasks have shown that nucleus reuniens disruption causes memory deficits⁶⁸, while silencing of medial prefrontal cortex reduces both behavioral performance and the magnitude of trajectory-dependent activity in CA1⁶⁹. These studies highlight the importance of trajectory-dependent activity to the performance of more difficult, hippocampal dependent memory tasks.

Along the same lines, the paragraph on page 24-25, lines 478-496, should emphasize that hippocampal dysfunction impairs performance of the discrete trial spatial memory tasks in which splitter cell activity predicted performance (e.g. Ferbinteanu, et al., 2003).

Another good suggestion. We have switched the order of this and the preceding paragraph to improve flow and have added the following:

“Finally, the results of Ferbinteanu & Shapiro²⁵ demonstrate that decreases in CA1 trajectory-coding correlate with impaired performance in a task with discrete trial structure. This suggests that the trajectory-dependent activity observed during hippocampal-independent tasks becomes necessary for proper memory-retrieval when task demands require higher levels of hippocampal engagement.”

Reviewer #3 (Remarks to the Author):

The authors did a thorough job in addressing my original concerns, which I think has enhanced the paper and increased its transparency. I have no additional comments and feel it is ready for publication and will be a very nice addition to the field.